# OIG-BENCH: A MULTI-AGENT ANNOTATED BENCHMARK FOR MULTIMODAL ONE-IMAGE GUIDES UNDERSTANDING

## ABSTRACT

Recent advances in Multimodal Large Language Models (MLLMs) have demonstrated impressive capabilities. However, evaluating their capacity for human-like understanding in **One-Image Guides** remains insufficiently explored. One-Image Guides are a visual format combining text, imagery, and symbols to present reorganized and structured information for easier comprehension, which are specifically designed for human viewing and inherently embody the characteristics of human perception and understanding. Here, we present **OIG-Bench**, a comprehensive benchmark focused on One-Image Guide understanding across diverse domains. To reduce the cost of manual annotation, we developed a semi-automated annotation pipeline in which multiple intelligent agents collaborate to generate preliminary image descriptions, assisting humans in constructing image–text pairs. With OIG-Bench, we have conducted comprehensive evaluation of 31 state-of-the-art MLLMs, including both proprietary and open-source models. The results show that Qwen2.5-VL-72B performs the best among the evaluated models, with an overall accuracy of 77%. Nevertheless, all models exhibit notable weaknesses in semantic understanding and logical reasoning, indicating that current MLLMs still struggle to accurately interpret complex visual-text relationships. In addition, we also demonstrate that the proposed multi-agent annotation system outperforms all MLLMs in image captioning, highlighting its potential as both a high-quality image description generator and a valuable tool for future dataset construction.

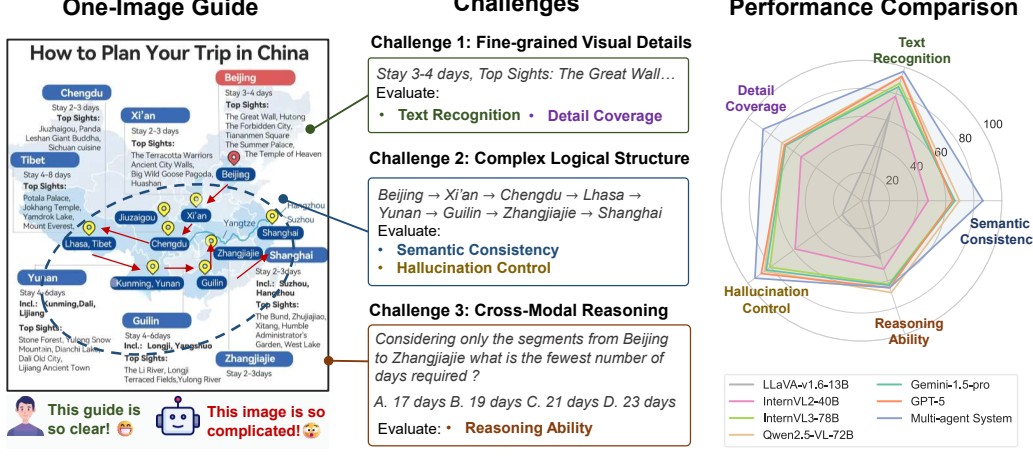

Figure 1: Left: An example of a One-Image Guide. Middle: Three major challenges for MLLMs in understanding One-Image Guides. Right: Results of six representative MLLMs and the proposed multi-agent annotation system. The left-skewed distribution indicates that MLLMs still perform poorly in logical reasoning and semantic understanding. In contrast, the multi-agent annotation system achieves the best overall performance.

# 1 INTRODUCTION

Multimodal Large Language Models (MLLMs) have recently achieved groundbreaking progress Li et al. (2023); Liu et al. (2023); Zhu et al. (2023); Wang et al. (2024), demonstrating the ability to process and integrate information from multiple modalities, including text, images, and audio. These models have demonstrated impressive performance in tasks such as image captioning Bucciarelli et al. (2024) and visual question answering (VQA) Liu et al. (2024b). With the continuous improvement of model capabilities, how to scientifically and comprehensively evaluate the multimodal understanding and reasoning ability of MLLM has become a key issue to promote further.

One of the ultimate objectives of MLLMs is to demonstrate human-like perceptual and cognitive abilities. Nevertheless, evaluating the human-likeness of MLLMs remains a significant challenge. To assess such capabilities, existing MLLM evaluation benchmarks often focus on complex scenarios, particularly those involving visually complex and information-rich images, such as infographics. Existing benchmarks for infographic understanding have made valuable contributions Mathew et al. (2022); Masry et al. (2022); Li et al. (2024) by targeting structured visual data, including charts and tables. However, these datasets typically feature regular layouts and fixed element positions, making them insufficient to fully assess a model's comprehensive reasoning and semantic understanding ability at a human-like level.

To bridge this gap, we focus on a visual information format that inherently embodies the characteristics of human-like perception and understanding: **One-Image Guide**. A One-Image Guide is a single image that presents rich task-relevant information in a compact, visually engaging format. (see Fig. 1 Left). It leverages human cognitive advantages such as parallel visual processing and pattern recognition, enabling rapid comprehension of the overall structure and efficient extraction of key information. While this format is highly accessible to humans, it poses significant challenges for MLLMs due to its fine-grained visual details, complex logical structure, and cross-modal reasoning (see Fig. 1 Middle). To achieve a complete and accurate understanding of one-image guides, MLLMs must possess fine-grained visual perception (e.g., text and details recognition) and spatial structure parsing capabilities (e.g., interpreting layouts, arrows, and segmented regions). These abilities must be effectively integrated to form coherent and contextually grounded semantic interpretations. As the One-Image Guide is optimized for human cognitive processing, the ability to comprehend it with human-like ease can serve as evidence of human-like visual understanding.

Here, we introduce OIG-Bench, a comprehensive benchmark for evaluating MLLMs on the understanding of one-image guides. OIG-Bench is a bilingual dataset containing 808 images spanning multiple domains. To construct this benchmark efficiently, we innovatively propose a semi-automated annotation pipeline based on a multi-agent collaboration framework. In this framework, multiple intelligent agents collaborate to generate detailed descriptions of one-image guides, which are then refined through human verification. This approach substantially reduces manual annotation effort while ensuring high-quality labels. Meanwhile, we also demonstrate that such multi-agent annotation system serves as an effective approach for high-quality image description generation (see Fig. 1 Right). Based on these annotated images, we further conduct extensive evaluations, encompassing 31 prominent MLLMs, including GPT-5, Qwen2.5-VL, and InternVL3. We design two evaluation tasks, including image description generation and VQA, to assess five key capabilities. Our evaluation shows that Qwen2.5-VL-72B achieves the best overall performance, but all models exhibit notable weaknesses in semantic consistency and reasoning. Through this evaluation, we not only benchmarked the current MLLMs but also provided key insights to drive the model to achieve more human-like perception and cognitive abilities in complex, information-intensive multimodal scenarios. Datasets are available at https://anonymous.4open.science/status/OIG-Bench-6628.

In summary, our main contributions are as follows:

(1) To the best of our knowledge, our proposed OIG-Bench is the first MLLM evaluation benchmark dedicated to one-image guide understanding.

(2) We introduce a multi-agent-based semi-automated benchmark construction method, which can serve as an effective image description generation tool to reduce manual annotation costs.

(3) We conduct a systematic evaluation of 31 mainstream MLLMs across five dimensions of one-image guide understanding. The results reveal that existing models still have substantial room for improvement in semantic understanding and logical reasoning.

## 2 RELATED WORKS

### 2.1 MULTIMODAL LARGE LANGUAGE MODELS

Multimodal Large Language Models (MLLMs) have rapidly advanced at the intersection of computer vision and natural language processing. Early systems used separate encoders and shallow fusion, exemplified by CLIPRadford et al. (2021). With the rise of large language models (LLMs) such as GPT-3Brown et al. (2020), visual encoders were integrated with generative language backbones to create general-purpose MLLMs. A typical architecture combines a pretrained vision encoder (e.g., CLIP-ViT, Swin TransformerLiu et al. (2021)), a modality alignment module (e.g., linear projection, Q-FormerLi et al. (2023)), and an LLM (e.g., LLaMATouvron et al. (2023), OPTZhang et al. (2022), GPT Brown et al. (2020)) for reasoning and generation.

Several influential MLLMs have been proposed in recent years. BLIP-2 Li et al. (2023) introduced a lightweight Q-Former to bridge frozen vision encoders and frozen LLMs, achieving strong performance with minimal training cost. MiniGPT-4 Zhu et al. (2023) demonstrated that aligning CLIP visual features with Vicuna's Team (2023) language space enables rich image-grounded conversations. LLaVA Liu et al. (2023) further improved instruction-following capabilities by fine-tuning on large-scale image–text instruction datasets. Commercial systems such as GPT-4o and Gemini have extended these capabilities to more modalities and complex reasoning tasks. Despite the rapid progress of MLLMs, systematic evaluation of their performance on text-rich, information-dense images remains insufficient, leaving a gap in understanding their true capabilities in such challenging scenarios.

### 2.2 MLLMS BENCHMARKS

The rapid growth of Multimodal Large Language Models (MLLMs) has led to numerous benchmarks for evaluating vision–language capabilities. Early evaluation datasets, such as VQAv2 Yang et al. (2022), COCO Caption Chen et al. (2015), and Flickr30k Entities Plummer et al. (2015), primarily targeted specific tasks like visual question answering, image captioning, and referring expression comprehension. With the emergence of instruction-following MLLMs, more comprehensive evaluation suites have been proposed to assess perception, reasoning, and knowledge-based abilities in a unified framework. For example, MMBench Liu et al. (2024b) evaluates models through carefully designed multiple-choice questions covering diverse multimodal skills, while MME Fu et al. (2024) provides large-scale, fine-grained assessments across perception, cognition, and reasoning dimensions.

While these benchmarks cover a broad spectrum of multimodal tasks, they may not fully capture the challenges posed by text-rich, information-dense images, which require accurate OCR, spatial layout understanding, and fine-grained multimodal reasoning. To address this, several specialized benchmarks have been developed. TextVQA Singh et al. (2019) focuses on natural scene images containing embedded text, evaluating models' abilities to read and reason over textual content. DocVQA Mathew et al. (2021) targets document images such as forms and invoices, emphasizing structured layout parsing. ChartQA Masry et al. (2022) assesses reasoning over charts and plots, requiring numerical understanding and visual–textual integration. Seed-Bench-2-PLUS further expands the evaluation scope to include text-rich images, charts, and diagrams. In the infographic domain, InfographicVQA Mathew et al. (2022) contains infographic-style images that combine visual elements, icons, and explanatory text, challenging models to understand across heterogeneous visual and textual components. In addition, reasoning-oriented benchmarks such as $M^3CoT$ Chen et al. (2024a) and CoMT Cheng et al. (2025) are designed to evaluate MLLMs' ability to perform multi-hop reasoning, also posing a challenge to existing MLLMs

## 3 OIG-BENCH

The entire data construction process is illustrated in Figure 2. We adopt a semi-automated approach to collect, filter, and annotate the dataset. In this section, we detail the semi-automated data processing pipeline, as well as the evaluation tasks and data analysis of our benchmark.

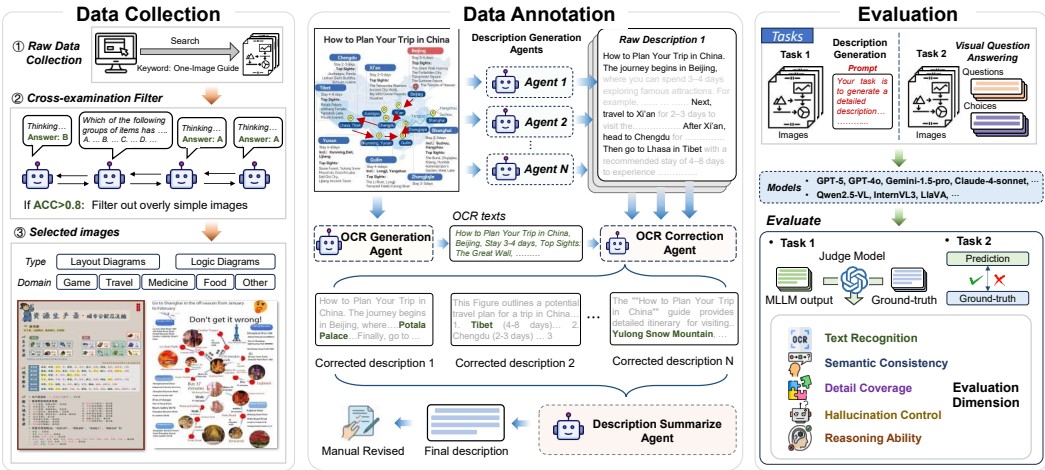

Figure 2: The One-Image Guide Data Processing Pipeline. We collect One-Image Guide images from the Internet and filter them using a cross-examination process with multiple MLLMs, where one model poses questions and others answer. Images with accuracy above 0.8 are considered too simple and removed. Next, we annotate the images using a multi-agent system, followed by manual verification. In the evaluation stage, we assess MLLMs on two tasks, namely description generation and VQA, covering five key capabilities.

## 3.1 SEMI-AUTOMATED DATA PROCESSING PIPELINE

### 3.1.1 DATA CURATION PROCESS

**Data Collection:** We collected One-Image Guides from multiple online platforms, including Red Note, Baidu, and Google, anonymizing all real names to ensure privacy. We categorize the collected images into two types: **layout diagrams** that present ordered or ranked information through tables, columns, or lists (e.g., game character tier rankings, recommended travel destination lists), and **logic diagrams** that illustrate processes, causal relationships, or conceptual connections (e.g., game character relationship maps, travel route maps). Examples can be found in Figure 6 in Appendix A.1.

**Data Filtration:** To ensure that the dataset remained both high-quality and sufficiently challenging, we employed a filtering process assisted by MLLMs. We adopted a mutual cross-examination strategy in which a subset of MLLMs generated questions for each image while the remaining models attempted to answer them. If the average accuracy exceeded 80%, the image was deemed too simple and removed. A final manual review was conducted to eliminate any remaining low-quality samples, resulting in a curated set of 808 images for subsequent annotation and evaluation. More details can be found in Appendix A.2.

### 3.1.2 MULTI-AGENT-BASED DESCRIPTION GENERATION

Manually creating accurate and information-rich textual descriptions for each image is not only time-consuming and labor-intensive, but also prone to inconsistencies in style and detail across annotators. To address these challenges, we design a multi-agent collaboration framework that integrates the complementary capabilities of multiple specialized agents.

**Description Generation Agent:** Given an input image, the process begins with several state-of-the-art multimodal large language models (MLLMs) acting as description generation agents. Each of these agents independently analyzes the visual content and produces an initial description, capturing salient objects, attributes, and relationships from its own perspective. Specifically, we employ GPT-4o, Claude-4-Sonnet, Gemini-1.5-Pro, and Qwen2.5-VL-78B as the description generation agents. This diversity in initial outputs helps mitigate the bias or omission that may occur when relying on a single model.

**OCR Generation Agent:** In parallel, the image is processed by an OCR generation agent, which is responsible for detecting and extracting any textual content embedded within the image, such as labels, annotations, or captions. This agent employs an OCR-specific model, PaddleOCR, rather than a large language model, as specialized OCR systems are generally more accurate and efficient for text recognition tasks.

**OCR Correction Agent:** Once both the initial descriptions and OCR results are obtained, an OCR correction agent refines each initial description by cross-referencing it with the extracted text. This process corrects transcription errors, fills in missing textual details, and ensures that all in-image text is faithfully represented in the description. We implement this agent using GPT-4.1, leveraging its strong language understanding and reasoning capabilities to integrate OCR outputs with visual descriptions effectively.

**Description Summarize Agent:** Finally, a description summarize agent aggregates all the corrected descriptions into a single, coherent, and comprehensive final description. This agent consolidates overlapping information, resolves inconsistencies across different sources, and ensures that the final output is logically structured and semantically complete. We implement this agent using GPT-4.1. By integrating the strengths of visual understanding, text recognition, and summarization, the multi-agent pipeline produces descriptions that are both accurate and information-rich, providing a robust foundation for subsequent annotation and evaluation tasks.

This multi-agent-based description generation system can serve as an effective tool for producing high-quality image descriptions by leveraging the complementary strengths of multiple specialized agents. Finally, all automatically generated descriptions are manually reviewed and corrected by human annotators, and the refined results are used as the final ground-truth annotations for the images. The prompts for each agent are provided in the Appendix A.8.

## 3.2 EVALUATION TASKS

To comprehensively assess the model's capabilities, we employ two evaluation tasks: Description Generation and Visual Question Answering (VQA).

(1) Description Generation. Given an image, the model is required to produce a coherent and semantically accurate textual description. The evaluation focuses on four core capabilities:

- **Text Recognition:** Measures the correctness of transcribed textual content from the visual input, reflecting the model's OCR performance.

- **Semantic Consistency:** Evaluates the alignment between the generated description and the ground truth in terms of factual correctness, logical coherence, and temporal or numerical accuracy.

- **Detail Coverage:** Assesses whether the generated description captures all relevant and salient details present in the image, including objects, attributes, and contextual information.

- **Hallucination Control:** Identifies instances where the model introduces content not supported by the visual input, indicating over-generation or fabrication.

For each aspect, we employ GPT-4.1 to compare the model-generated description with the human-verified ground truth and assign a score from 0 to 1 on a six-level rating scale (0.0, 0.2, 0.4, 0.6, 0.8, 1.0). To validate the reliability of GPT-4.1 scoring, we additionally conducted human evaluations on several representative models. As shown in Table 7 and Figure 8 in Appendix A.4, the GPT-4.1 scores exhibit strong consistency with human scores, indicating the robustness and accuracy of the automated evaluation.

(2) VQA: In the VQA task, the model is required to answer natural language questions based on the given visual input. This task is specifically designed to evaluate the model's **Reasoning Ability**. To construct the question set, we leverage the ground-truth descriptions in our dataset and employ GPT-4.1 to automatically generate 3-5 questions for each description. The questions are intentionally constructed to require multi-step reasoning, such as combining multiple pieces of information, performing temporal or numerical inference, or integrating contextual cues. To improve robustness, we adopted the CircularEval strategy used in MMBench Liu et al. (2024b), which evaluates each question across all circular permutations of the answer options.

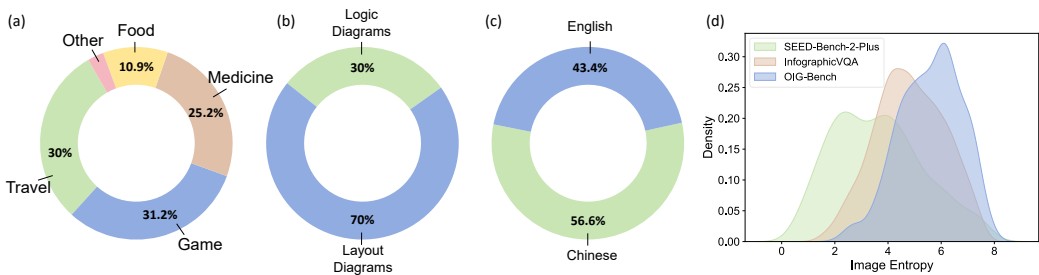

Figure 3: Statistics of OIG-Bench. (a) Domain distribution: Game (48.8%), Travel (28.7%), Food (9.1%), Medicine (8.9%), and Other. (b) Diagram type distribution: Layout Diagrams (70.5%) and Logic Diagrams (29.5%). (c) Kernel density estimation of image entropy, comparing OIG-Bench with SEED-Bench-2-Plus and InfographicVQA.

### 3.3 DATASET ANALYSIS

Our proposed OIG-Bench contains a total of 808 one-image guide images accompanied by 2,800 questions. The dataset spans multiple domains, including game, travel, medicine, and food, with the game and travel domains being the most represented, accounting for 31.2% and 30% of all images (Fig. 3(a)), respectively. Each question is provided with four options. The average question length is approximately 67.5 words, while the average option length is around 74.9 words. In our benchmark, the OIG images can be broadly classified into two categories, with logic diagrams accounting for 30% and layout diagrams for 70% of the dataset (Fig. 3(b)). Furthermore, the dataset is bilingual, with Chinese images accounting for 56.6% and English images for 43.4% (Fig. 3(c)).

To quantitatively assess the visual complexity of our dataset, we compute the image entropy for each image. Image entropy measures the amount of information contained in an image and is defined as:

$$H = - \sum_{i=0}^{L-1} p_i \log_2 p_i, \tag{1}$$

where $L$ denotes the number of possible pixel intensity levels, and $p_i$ represents the probability of intensity level $i$ in the image. Higher entropy values indicate greater variability and complexity in pixel distribution, which typically correspond to richer visual content.

We compare the image entropy of OIG-Bench with SEED-Bench-2-Plus and InfographicVQA. As shown in Figure 3(d), OIG-Bench has a higher average entropy (5.8) than InfographicVQA (4.8) and SEED-Bench-2-Plus (3.2). This substantial difference suggests that OIG-Bench images are more visually complex and information-dense, thereby posing greater challenges for multimodal large language models.

## 4 EXPERIMENTS

### 4.1 EXPERIMENT SETUPS

We evaluate a total of 31 models, including 7 leading proprietary (closed-source) models and 24 SOTA open-source models. The proprietary models include GPT5, GPT-4o, Claude-4-sonnet, Claude-3-7-sonnet, and Gemini-1.5-pro. The open-source models include InternVL3.5 Wang et al. (2025), InternVL3 Zhu et al. (2025), InterVL2.5 Chen et al. (2024b), InterVL2 Chen et al. (2024c), Qwen2.5-VL Bai et al. (2025), Qwen2-VL Wang et al. (2024), Yi-VL Young et al. (2024), LLaVA-v1.6 Liu et al. (2024a), and LLaVA-v1.5 Liu et al. (2024a), covering various model scales from 6B to 78B. To ensure fairness and comparability, all models are prompted using the same instruction template and set with a temperature of 0. The maximum completion token length is fixed at 2048. For subsequent few-shot experiments, we reserve two images from each domain, with the remaining images used exclusively for evaluation. The initial descriptions from the multi-agent annotation sys-

Table 1: Performance comparison of proprietary and open-source MLLMs on OIG-Bench. The overall score (%) is calculated as the mean of the scores (%) across five dimensions. The best and second-best results among evaluated MLLMs are highlighted in bold and underlined, respectively.

| Model | Scale | Overall | Description Generation | | | | VQA |
| --- | --- | --- | --- | --- | --- | --- | --- |
| | | | Semantic Consistency | Text Recognition | Detail Coverage | Hallucination Control | Reasoning Ability |
| Open-Source MLLMs | | | | | | | |
| LLaVA-1.5 | 7B | 29.16 | 6.10 | 71.69 | 8.49 | 26.19 | 31.33 |
| | 13B | 27.97 | 4.11 | 76.93 | 7.45 | 13.95 | 35.42 |
| LLaVA-1.6 | 7B | 22.31-1.29 | 2.79 | 53.11 | 4.78 | 11.44 | 32.96 |
| | 13B | 27.04 | 5.00 | 66.40 | 8.74 | 17.12 | 37.91 |
| Yi-VL | 6B | 33.04 | 20.42 | 75.35 | 10.54 | 18.49 | 38.41 |
| | 34B | 35.24 | 24.46 | 76.79 | 9.68 | 19.55 | 41.72 |
| InternVL2 | 26B | 54.99 | 46.71 | 75.50 | 52.75 | 58.78 | 44.19 |
| | 40B | 55.34 | 47.52 | 78.11 | 53.20 | 58.42 | 41.42 |
| InternVL2.5 | 8B | 59.28 | 53.65 | 84.60 | 51.03 | 61.15 | 47.99 |
| | 26B | 61.37 | 54.96 | 85.30 | 57.70 | 63.06 | 47.82 |
| | 38B | 63.78 | 55.14 | 87.61 | 57.60 | 66.98 | 51.59 |
| | 78B | 66.55 | 59.11 | 87.90 | 57.53 | 73.85 | 54.35 |
| InternVL3 | 9B | 62.37 | 54.37 | 83.29 | 58.69 | 68.74 | 48.75 |
| | 14B | 64.91 | 57.70 | 84.40 | 61.04 | 73.59 | 49.82 |
| | 38B | 66.84 | 58.60 | 86.76 | 66.53 | 74.55 | 47.76 |
| | 78B | 71.38 | 65.69 | 87.99 | 67.13 | 80.54 | 57.55 |
| InternVL3.5 | 8B | 62.60 | 54.37 | 83.29 | 58.69 | 68.74 | 47.88 |
| | 14B | 68.59 | 60.00 | 83.79 | 67.07 | 74.90 | 57.19 |
| | 38B | 68.58 | 60.95 | 84.29 | 66.30 | 75.62 | 55.72 |
| Qwen2-VL | 7B | 66.50 | 56.89 | 91.04 | 57.97 | 80.32 | 46.28 |
| | 72B | 71.74 | 64.11 | 90.97 | 64.15 | 83.57 | 53.90 |
| Qwen2.5-VL | 7B | 69.77 | 59.73 | 91.08 | 60.68 | 82.39 | 54.98 |
| | 32B | 75.09 | 69.84 | 92.25 | **70.74** | 80.32 | 62.32 |
| | 72B | **76.52** | **70.48** | 93.02 | 70.07 | 85.42 | **63.62** |
| Proprietary MLLMs | | | | | | | |
| Gemini-1.5-pro | – | 72.07 | 66.54 | 85.31 | 66.98 | 83.93 | 55.62 |
| Gemini-2.5-pro | – | 72.30 | 67.12 | 89.18 | 66.32 | 83.07 | 55.81 |
| Claude-3-7-sonnet | – | 64.27 | 58.69 | 75.35 | 63.79 | 68.67 | 56.88 |
| Claude-4-sonnet | – | 66.24 | 61.99 | 76.20 | 65.93 | 75.52 | 53.55 |
| GPT-4o | – | 70.14 | 63.77 | 77.61 | 65.38 | 87.95 | 55.98 |
| o3 | – | 71.15 | 63.36 | 84.15 | 65.14 | 85.24 | 57.82 |
| GPT-5 | – | 73.44 | 64.96 | **93.05** | 67.70 | **88.24** | 55.26 |
| Multi-agent | – | 84.99 | 86.49 | 96.85 | 86.67 | 93.83 | 61.07 |

tem are included in the evaluation, but as it cannot perform VQA directly, we combine its generated description with the question and use GPT-4.1 to produce the answer for fair comparison.

## 4.2 OVERALL PERFORMANCE OF OPEN- AND CLOSED-SOURCE MODELS

Table 1 summarizes the results for the description generation and VQA tasks. We have the following observations:

**(1) Performance gap between open- and closed-source MLLM is narrowing.** Overall, closed-source models generally perform better than open-source models, particularly in hallucination control. Nevertheless, several open-source models are able to match or even surpass closed-source models. In particular, Qwen2.5-VL-72B achieves the highest overall score among all evaluated models, outperforming the best closed-source model, GPT-5, by 2.6%. Qwen2.5-VL-72B also excels in key metrics such as semantic consistency and reasoning ability, highlighting its advanced capability in interpreting complex visual content.

Table 3: Ablation study of the multi-agent annotation system.

| Model | Overall | Semantic Consistency | Text Recognition Accuracy | Detail Coverage | Hallucination Detection |
|---|---|---|---|---|---|
| Qwen2.5-VL-72B | 79.75 | 70.48 | 93.02 | 70.07 | 85.42 |
| Multi-agent System w/o OCR | 86.09 | 77.41 | 93.36 | 84.41 | 89.19 |
| Multi-agent System w/o Summarize | 80.53 | 71.23 | 95.41 | 70.51 | 84.97 |
| Multi-agent System (Claude-4-Sonnet) | 90.38 | 85.79 | 96.14 | 85.98 | 93.61 |
| Multi-agent System (Gemini-2.5-Pro) | 90.20 | 85.34 | 97.03 | 84.13 | 94.31 |
| Multi-agent System (GPT-4.1) | 90.96 | 86.49 | 96.85 | 86.67 | 93.83 |

Table 4: Inference time of single-model and multi-agent settings on OIG-Bench.

| Model | Agents | Inference time per image (s) | Overall score |
|---|---|---|---|
| Qwen2.5-VL-72B | – | 26.34 | 77.56 |
| GPT-4o | – | 5.13 | 71.93 |
| Multi-agent | GPT-4o, Claude-4, Gemini-1.5-pro, Qwen2.5-VL-72B | 66.39 | **86.24** |
| Multi-agent | GPT-4o, Claude-4, Gemini-1.5-pro | 38.34 | 82.49 |

**(2) MLLMs exhibit strong text recognition ability but struggle with semantic consistency.** Most models reach around 80% accuracy in text recognition, but their performance on semantic consistency drops to about 60%.

**(3) Semantic understanding capability improves as model scale increases.** For example, in the Qwen family from 7B to 72B, we see consistent improvements across five capabilities, with the largest gains in semantic consistency and hallucination control. Nevertheless, several metrics, particularly text recognition and detail coverage, tend to plateau as model size grows, likely due to the fixed capacity of the vision encoder, which limits the extraction of fine-grained visual information. For instance, InternVL2.5-26B, InternVL2.5-38B, and InternVL2.5-78B all employ a 6B-parameter ViT, yet their performance in text recognition and detail coverage shows little difference.

**(4) Weak semantic understanding substantially hinders logical reasoning.** We further analyzed the relationship between semantic understanding and logical reasoning by calculating the semantic consistency scores of images with correct and incorrect VQA answers in GPT-4o, GPT-5, and Qwen2.5-VL-72B. The results shown in Table 2 indicate that all three models exhibit significantly higher semantic consistency scores (p-values<0.05 in all cases) when VQA answers are correct compared to when they are incorrect. When models are provided with the ground-truth text, their VQA performance increases substantially (Table 14). These results indicate a significant correlation between semantic understanding and logical reasoning.

Table 2: Semantic consistency scores for correct and incorrect VQA answers across different models, along with corresponding $p$-values.

| Model | Correct VQA answer | Incorrect VQA answer | $p$-value |
|---|---|---|---|
| GPT-4o | 64.77 | 59.49 | $6 \times 10^{-5}$ |
| GPT-5 | 65.82 | 53.04 | $1 \times 10^{-9}$ |
| Qwen2.5-VL-72B | 72.14 | 65.75 | $2 \times 10^{-5}$ |

### 4.3 PERFORMANCE OF THE MULTI-AGENT ANNOTATION SYSTEM

The proposed multi-agent annotation system yields the strongest description-generation results in our benchmark. It achieves 86.4% in semantic consistency, 96.8% in text recognition, 86.6% in detail coverage, and 93.8% in hallucination control, exceeding any single model baseline on each metric.

The gains can be attributed to the complementary strengths of multiple specialized agents, which can reduce hallucinations while improving coverage of fine-grained details in text-rich scenes. To

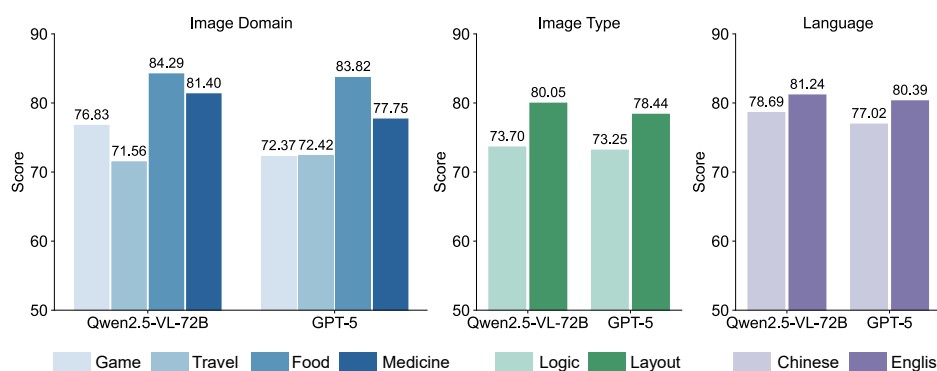

Figure 4: Performance comparison of two representative MLLMs across different image domains, types, and languages

validate the necessity of each component, we conducted a series of ablation studies, as shown in Table 3. When removing the OCR module (Multi-agent System w/o OCR), the text recognition accuracy drops considerably, and semantic consistency also degrades. This is because inconsistent or incorrect OCR outputs across agents lead to misaligned textual references, ultimately harming the global semantic alignment in the final annotations. When removing the summarization module and relying solely on a single model (Qwen2.5-VL-72B) with OCR correction (Multi-agent System w/o Summarize), we observe that although the text recognition accuracy improves, the scores for semantic consistency and hallucination detection drop significantly. This indicates that the summarization stage is essential for mitigating model-specific biases and consolidating globally coherent descriptions that go beyond the viewpoint of any single model. Finally, we replace the original agents with alternative models: Claude-4-Sonnet and Gemini-2.5-Pro. The resulting annotations remain consistently stronger than those produced by any single best-performing MLLM, demonstrating the superiority of the proposed multi-agent pipeline.

Notably, the multi-agent system involves invoking several large-scale model agents (GPT-4o, Claude-4, Gemini-1.5-pro, and Qwen2.5-VL-72B), inevitably introducing higher latency and computational costs. In particular, Qwen2.5-VL-72B that we deploy locally delivers the best performance but also incurs the longest inference time, averaging 26 seconds per image as shown in Table 4. In contrast, GPT-4o, accessed via its official API, requires only 5 seconds. The complete multi-agent annotation system takes 66 seconds per image, which is considerably longer than that of a single-model setup. To mitigate this, we removed Qwen2.5-VL-72B from the description generation agent, reducing the inference time while maintaining an overall score that still surpasses any single-model baseline. This highlights an inherent trade-off between performance and efficiency in multi-agent designs.

### 4.4 Performance across Different Image Domains, Types, and Languages

Figure 4 presents the performance of the evaluated models across different image domains, types, and languages. The results indicate that model scores vary substantially across domains: models generally achieve higher scores in the medical and food domains, while consistently performing worse in the gaming and travel domains. This disparity may be attributed to the open-world, natural, and icon-rich scenes that are typical of gaming and travel images, which pose greater challenges for accurate visual understanding and description generation. When comparing image types, we observe that logic diagrams are often more difficult than layout diagrams. We further demonstrate the types of errors in the logic diagram in Appendix Table 14. The results indicate that MLLM performs particularly poorly in interpreting arrows, revealing clear weaknesses in directional understanding and spatial reasoning. Regarding language differences, our analysis shows that Chinese-language images are generally more difficult. Across both languages, Qwen2.5-VL-72B consistently achieves the strongest performance, with a notably large margin in Chinese semantic consistency (Figure 10). One likely reason is that the Qwen2.5-VL family has been trained on a substantially larger volume of Chinese multimodal data, which enhances its robustness in Chinese multimodal understanding tasks.

Table 5: Overall results of different prompts on OIG-Bench.

| Model | Prompt | Overall | Description Generation | | | | VQA |
|---|---|---|---|---|---|---|---|
| | | | Semantic Consistency | Text Recognition Accuracy | Detail Coverage | Hallucination Detection | Reasoning Ability |
| GPT-4o | Base | 71.93 | 63.77 | 77.61 | 65.38 | 87.95 | 64.93 |
| | CoT | 71.26 (-0.67) | 63.59 (-0.18) | 76.14 (-1.47) | 65.17 (-0.21) | 84.31 (-3.64) | 67.11 (+2.18) |
| | 1-shot | 71.28 (-0.65) | 63.12 (-0.65) | 77.49 (-0.12) | 64.54 (-0.84) | 86.14 (-1.81) | 65.13 (+0.20) |
| | 2-shot | 71.40 (-0.53) | 63.45 (-0.32) | 77.31 (-0.30) | 64.37 (-1.01) | 86.37 (-1.58) | 65.49 (+0.56) |
| Qwen2.5-VL-72B | Base | 77.56 | 70.48 | 93.02 | 70.07 | 85.42 | 68.83 |
| | CoT | 77.15 (-0.41) | 69.17 (-1.31) | 93.12 (+0.10) | 69.73 (-0.34) | 84.02 (-1.40) | 69.71 (+0.88) |
| | 1-shot | 77.00 (-0.56) | 69.10 (-1.38) | 93.06 (+0.04) | 69.01 (-1.06) | 84.82 (-0.60) | 68.97 (+0.14) |
| | 2-shot | 76.57 (-0.99) | 68.45 (-2.03) | 92.25 (-0.77) | 68.39 (-1.68) | 84.62 (-0.80) | 69.12 (+0.29) |

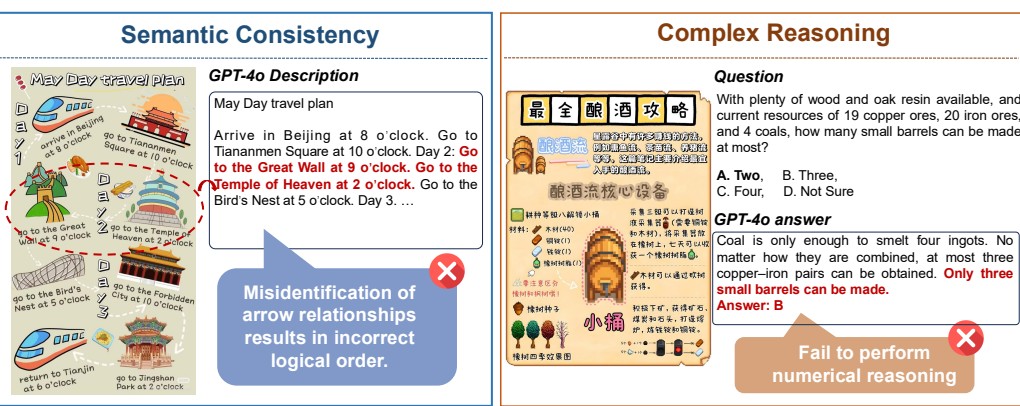

Figure 5: Case analysis of model errors.

## 4.5 ANALYSIS ON DIFFERENT PROMPT SKILLS

Table 5 shows the impact of Base, CoT, and few-shot prompts on image description generation and VQA performance. We observe that CoT and few-shot prompts yield small gains in VQA reasoning, yet they reduce description quality. The largest declines occur in hallucination control and detail coverage. One possible explanation is that CoT prompts tend to elicit more exploratory outputs, which can be beneficial for arithmetic or aggregation questions in VQA but may harm the quality of free-form descriptions by introducing additional hallucinated content.

## 4.6 ERROR ANALYSIS

This subsection provides a case study for analyzing two major weaknesses of current models: semantic consistency and complex reasoning. One of the most common errors arises from incorrect identification of arrow relationships, which leads to faulty logical sequencing. This issue is particularly pronounced when the two entities connected by an arrow are spatially close in the image. As illustrated in Figure 5, GPT-4o reverses the visiting order between the Great Wall and the Temple of Heaven. Another frequent error involves the inability to handle complex reasoning tasks. The model demonstrates a pronounced weakness in multi-step inference, particularly in scenarios involving arithmetic reasoning. Overall, these observations not only highlight the current limitations of the models but also inform potential improvements.

## 5 DISCUSSION

In this work, we present OIG-Bench, the first benchmark specifically designed for evaluating One-Image Guide understanding in MLLMs. We propose a semi-automated benchmark construction method and systematically evaluate 31 MLLMs. Experimental results show that current MLLMs still exhibit limitations in semantic understanding and logical reasoning. We hope OIG-Bench to provide a challenging, realistic platform that drives MLLMs toward human-level understanding of complex visual information.

## ETHICS STATEMENT

The study does not involve human subjects, and all datasets used are either publicly available or released by us under licenses that permit free usage for research purposes. We have taken care to ensure that the released datasets do not contain personally identifiable information and that data processing steps comply with relevant privacy and security regulations. The proposed methodologies and applications are not intended to produce harmful insights or outcomes, and we have evaluated potential risks related to bias, fairness, and discrimination. No conflicts of interest or sponsorship influence the reported results. All experiments and analyses follow established research integrity practices, and documentation is provided to facilitate transparency and reproducibility.

## REPRODUCIBILITY STATEMENT

The codes and data used in this study are publicly available at https://anonymous.4open.science/status/OIG-Bench-6628.

- **Dataset Usage**
    - This paper relies on one or more datasets (yes)
    - A motivation is given for why the experiments are conducted on the selected datasets (yes)
    - All novel datasets introduced in this paper are publicly available with a license that allows free usage for research purposes (yes)
    - All datasets drawn from the existing literature (potentially including authors' own previously published work) are accompanied by appropriate citations (yes)
    - All datasets drawn from the existing literature (potentially including authors' own previously published work) are publicly available (yes)

- **Computational Experiments**
    - This paper includes computational experiments. (yes)
    - This paper states the number and range of values tried per (hyper-) parameter during development of the paper, along with the criterion used for selecting the final parameter setting. (yes)
    - All source code required for conducting and analyzing the experiments are publicly available with a license that allows free usage for research purposes (yes)
    - All source code implementing new methods have comments detailing the implementation, with references to the paper where each step comes from (yes)
    - If an algorithm depends on randomness, then the method used for setting seeds is described in a way sufficient to allow replication of results (yes)
    - This paper formally describes evaluation metrics used and explains the motivation for choosing these metrics (yes)

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

# A APPENDIX

## A.1 DETAILS OF OIG-BENCH

OIG-Bench is a bilingual evaluation dataset of One-Image Guides, containing both English and Chinese samples. It primarily covers four domains: travel, gaming, food, and medical. Figure 6 presents representative examples from OIG-Bench,

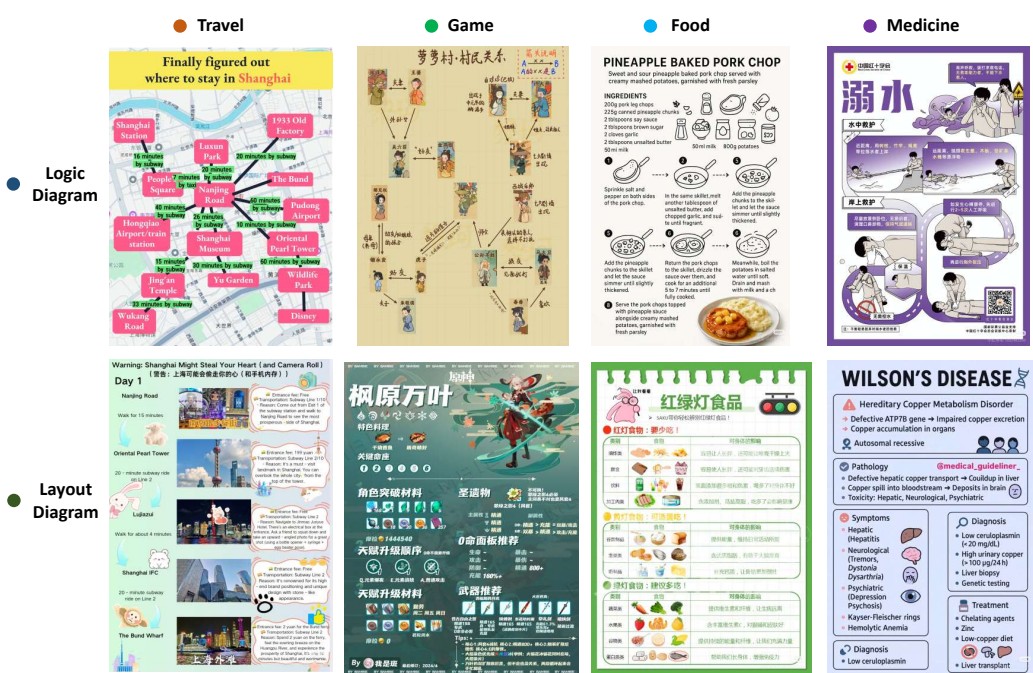

Figure 6: OIG-Bench examples sampled from each domain.

## A.2 DETAILS OF DATA FILTERING

We first collected 2,891 images from the internet using "one-image guide" as the primary search keyword. In the first stage, we performed a coarse manual screening to remove images that did not conform to the definition of a One-Image Guide, such as those lacking a clear integration of text, imagery, and symbols. This step resulted in 1,982 remaining images.

In the second stage, we applied an automated cross-validation system involving four advanced MLLMs: GPT-4o, Gemini-1.5-Pro, Claude-4, and Qwen2.5-VL-72B. The process was conducted in a round-robin manner. In each round, one model generated a multiple-choice question based on the content of a One-Image Guide, and the remaining models answered it. This procedure continued until each model had served as the question generator once. Images with an overall answering accuracy greater than 80% across all rounds were considered overly simple and thus excluded from the benchmark. This filtering step reduced the dataset to 1,023 images.

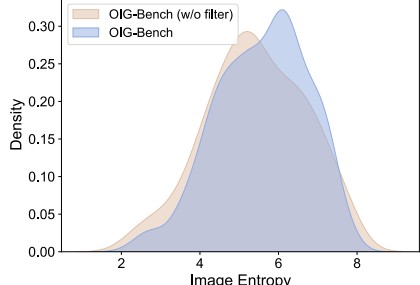

Figure 7: Kernel density estimation of image entropy of OIG-Bench before and after cross-validation filtering.

Finally, we conducted a fine-grained manual inspection to remove images with excessively low resolution or overly simple content. After this final filtering stage, 808 high-quality One-Image Guide images were retained for the benchmark. To validate the effective-

ness of the proposed cross-validation system, we analyzed and compared the image entropy distributions before and after this filtering stage. As shown in Figure 7, the post-filtering entropy distribution exhibits a clear rightward shift relative to the pre-filtering distribution, indicating that the filtering process successfully removed overly simple images while retaining those with richer informational content.

## A.3 DETAILS OF EVALUATED MLLMS

Table 6 provides an overview of the open-source MLLMs, highlighting differences in their architectures and parameters.

Table 6: The architecture and size of different models.

| Model | Scale | Vision Part | Language Part |
|---|---|---|---|
| LLaVA-1.5 | 7B | CLIP ViT-L/14@336px | Vicuna-7B |
| | 13B | CLIP ViT-L/14@336px | Vicuna-13B |
| LLaVA-1.6 | 7B | CLIP ViT-L/14@336px | Vicuna-7B |
| | 13B | CLIP ViT-L/14@336px | Vicuna-13B |
| Yi-VL | 6B | CLIP VIT-H/14 | Yi-6B-Chat |
| | 34B | CLIP VIT-H/14 | Yi-34B-Chat |
| Qwen2-VL | 7B | DFN(ViT-625M) | Qwen2-7B |
| | 72B | DFN(ViT-625M) | Qwen2-78B |
| Qwen2.5-VL | 7B | - | Qwen2.5-7B |
| | 32B | - | Qwen2.5-32B |
| | 72B | - | Qwen2.5-72B |
| InternVL2 | 26B | InternViT-6B-448px-V1-5 | Internlm2-chat-20b |
| | 40B | InternViT-6B-448px-V1-5 | Nous-Hermes-2-Yi-34B |
| InternVL2.5 | 8B | InternViT-300M-448px-V2_5 | Internlm2_5-7b-chat |
| | 26B | InternViT-6B-448px-V2_5 | Internlm2_5-20b-chat |
| | 38B | InternViT-6B-448px-V2_5 | Qwen2.5-32B-Instruct |
| | 78B | InternViT-6B-448px-V2_5 | Qwen2.5-72B-Instruct |
| InternVL3 | 9B | InternViT-300M-448px-V2_5 | Internlm3-8b-instruct |
| | 14B | InternViT-300M-448px-V2_5 | Qwen2.5-14B |
| | 38B | InternViT-6B-448px-V2_5 | Qwen2.5-32B |
| | 78B | InternViT-6B-448px-V2_5 | Qwen2.5-72B |
| InternVL3.5 | 8B | InternViT-300M-448px-V2_5 | Qwen3-8B |
| | 14B | InternViT-300M-448px-V2_5 | Qwen3-14B |
| | 38B | InternViT-6B-448px-V2_5 | Qwen3-32B |

## A.4 ANALYSIS OF AUTOMATED EVALUATION

In this study, we employed GPT-4.1 to automatically evaluate the accuracy of descriptions generated by MLLMs across five assessment dimensions, using human-annotated descriptions as references. To verify the reliability of the automatic scoring, we additionally conducted human evaluations on two representative models: GPT-5 and Qwen2.5-VL-72B. Each image was independently scored by three human annotators, and the average score was used for comparison. As shown in Table 7, the human scores and the GPT-4.1 automatic scores exhibit strong alignment. Figure 8 further illustrates the comparison of overall scores for each image, where the Pearson correlation coefficients between human and GPT-4.1 scores are 0.93 for Qwen2.5-VL-72B and 0.94 for GPT-5, confirming that GPT-4.1 can serve as a reliable proxy for evaluation in this benchmark.

Furthermore, to further validate the robustness of our evaluation, we conducted additional experiments using two extra judge models: Gemini 2.5 Pro and Claude-4-Sonnet. The evaluation follows the exact evaluation pipeline as the GPT-4.1. Results are shown in Table 8 and 9. The

Table 7: Comparison between GPT-4.1 automatic scoring and human scoring.

| Model | Score | Semantic Consistency | Text Recognition Accuracy | Detail Coverage | Hallucination Detection |
|---|---|---|---|---|---|
| Qwen2.5-VL-72B | GPT-4.1 | 70.48 | 93.02 | 70.07 | 85.42 |
| Qwen2.5-VL-72B | Huamn | 69.37 | 95.21 | 71.69 | 86.97 |
| GPT-5 | GPT-4.1 | 64.96 | 93.05 | 67.70 | 88.24 |
| GPT-5 | Human | 62.15 | 94.68 | 69.13 | 88.37 |

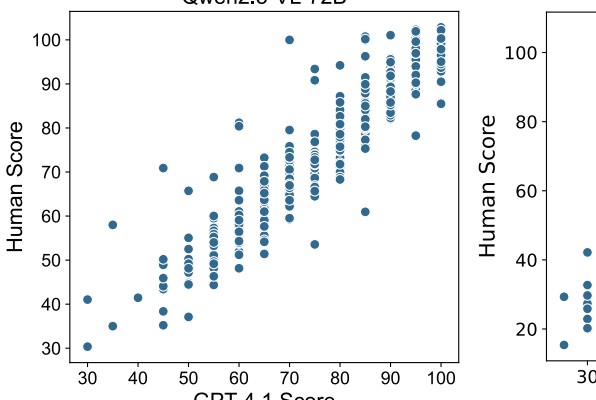 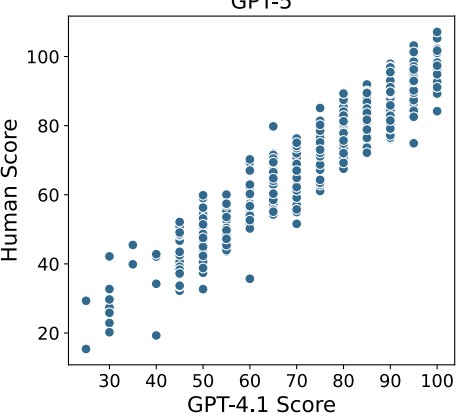

Figure 8: Correlation between GPT-4.1 and human overall scores.

Spearman correlation for the rank of these models between the GPT-4.1-based OIG-Bench and the Gemini-2.5-pro-based OIG-Bench is 0.884 with p-value<0.05, and between the GPT-4.1-based OIG-Bench and the Claude-4-Sonnet-based OIG-Bench is 0.903 with p-value<0.05. This consistency reflects that the inherent bias introduced by GPT-4.1 in OIG-Bench is not significant.

### A.5 PERFORMANCE ACROSS DIFFERENT IMAGE PIXELS

Figure 9 (left) presents the pixel distribution of images in OIG-Bench. We divided all images into three categories: low-resolution images with a total pixel count below 1.5M, medium-resolution images between 1.5M and 2.5M, and high-resolution images above 2.5M. Among these, medium-resolution images are the most common, accounting for 60.2% of the dataset. Figure 9 (right) shows the performance of Qwen2.5-VL-72B across the three resolution categories. We observe that as resolution increases, the model's scores on semantic consistency and detail coverage decrease noticeably, suggesting that higher-resolution images may contain more complex visual information and finer details, which increase the difficulty of accurate description generation. In contrast, text recognition and hallucination control do not exhibit a performance drop with increasing resolution, indicating that these aspects are less sensitive to image resolution within the tested range.

### A.6 DETAILED PERFORMANCE ACROSS DIFFERENT LANGUAGES

Figure 10 presents the performance of Qwen2.5-vl-72B and GPT-5 across four evaluation dimensions of different image languages.

### A.7 PERFORMANCE COMPARISON ON INFOGRAPHICVQA

To more intuitively demonstrate the unique value of OIG-Bench, we additionally evaluate several representative models on both the image description and VQA tasks on InfographicVQA. We sampled a subset of InfographicVQA containing 300 images, and applied the same annotation, verification, and evaluation procedures as in OIG-Bench. As shown in Table 10, for the same model,

Table 8: Performance comparison of proprietary and open-source MLLMs on OIG-Bench. All results are evaluated using **Gemini-2.5-Pro** as the judge model.

| Model | Overall | Semantic Consistency | Text Recognition Accuracy | Detail Coverage | Hallucination Detection |
|---|---|---|---|---|---|
| InternVL3-78B | 64.26 | 45.24 | 84.12 | 55.35 | 72.34 |
| Qwen2-VL-72B | 64.52 | 46.75 | 87.88 | 52.84 | 72.60 |
| Qwen2.5-VL-7B | 62.99 | 40.90 | 88.89 | 49.57 | 70.62 |
| Qwen2.5-VL-32B | 68.41 | 53.05 | **89.32** | **59.86** | 71.41 |
| Qwen2.5-VL-72B | **70.13** | **58.17** | 85.71 | 58.75 | 77.90 |
| Gemini-1.5-pro | 64.22 | 50.27 | 80.94 | 49.97 | 74.12 |
| Gemini-2.5-pro | 64.22 | 52.23 | 81.22 | 52.19 | 74.23 |
| GPT-4o | 67.04 | 50.67 | 83.43 | 53.67 | **82.38** |
| o3 | 65.86 | 53.41 | 81.67 | 50.79 | 77.57 |
| GPT-5 | 67.51 | 51.75 | 86.12 | 52.13 | 80.03 |

Table 9: Performance comparison of proprietary and open-source MLLMs on OIG-Bench. All results are evaluated using **Claude-4-Sonnet** as the judge model.

| Model | Overall | Semantic Consistency | Text Recognition Accuracy | Detail Coverage | Hallucination Detection |
|---|---|---|---|---|---|
| InternVL3-78B | 66.94 | 58.21 | 77.23 | 71.35 | 60.98 |
| Qwen2-VL-72B | 67.54 | 57.36 | 78.64 | 67.69 | 66.45 |
| Qwen2.5-VL-7B | 64.34 | 51.85 | 78.83 | 63.12 | 63.56 |
| Qwen2.5-VL-32B | 71.32 | 64.60 | 80.84 | **74.46** | 65.37 |
| Qwen2.5-VL-72B | **77.45** | **72.27** | **84.88** | 74.28 | 80.35 |
| Gemini-1.5-pro | 67.38 | 60.72 | 76.66 | 62.37 | 71.75 |
| Gemini-2.5-pro | 73.27 | 66.38 | 79.64 | 70.96 | 78.11 |
| GPT-4o | 70.53 | 63.17 | 67.88 | 66.71 | **82.38** |
| o3 | 74.51 | 64.12 | 78.98 | 71.97 | 80.97 |
| GPT-5 | 71.77 | 63.44 | 80.60 | 63.78 | 79.25 |

performance on OIG-Bench is consistently lower across all evaluation dimensions, with particularly large gaps in semantic consistency and logical reasoning. This quantifies the greater difficulty of OIG-Bench, which contains more diverse layouts, denser text regions, and richer multimodal reasoning requirements than InfographicVQA.

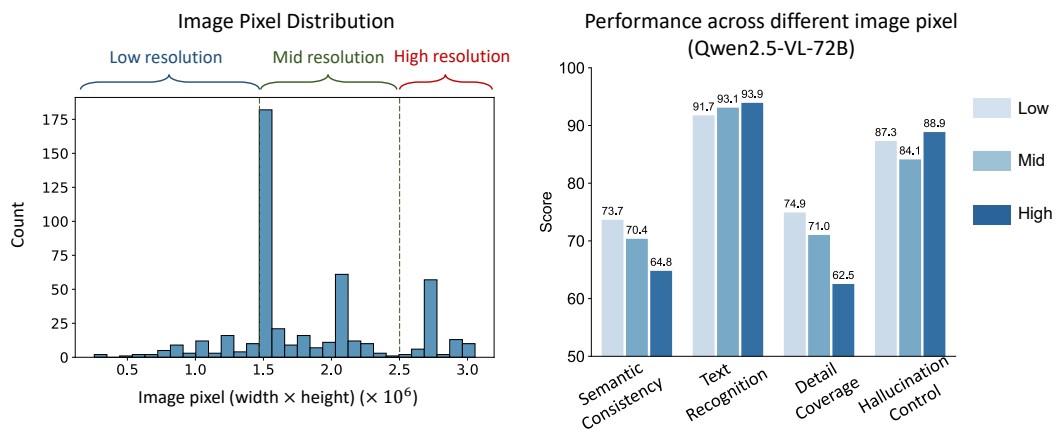

Figure 9: Left: The image pixel distribution of OIG-Bench. Right: Performance of Qwen2.5-VL-72B across different image resolutions.

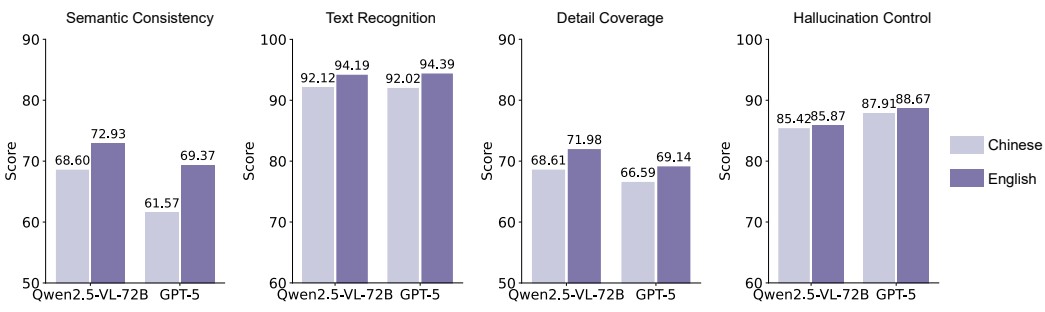

Figure 10: Performance across four dimensions of different image languages.

## A.8 PROMPT TEMPLATES

---

**Prompt to generate description of One-Image Guide (Description Generation Agent).**

**User:** <Image>. Your task is to generate a detailed description for a 'One-Image Guide'. Clearly convey the content of the guide by following these guidelines:

1. Carefully read and understand all the textual information in the guide image, ensuring you grasp every step and key point.

2. Present each step in a detailed and well-organized manner, following the logical sequence of the guide.

3. Use clear and easy-to-understand language. Only describe the content shown in the image; do not include any information that is not mentioned in the image, and do not provide additional introductions.

4. Ensure the description is complete and contains all necessary information.

---

**Prompt for OCR correction of generated descriptions (OCR Correction Agent).**

**User:** Your task is to revise the large language model's description of a 'One-Image Guide' based on the OCR model's extracted text.

Below is the large language model's output description:

<LLM Description>

---

Table 10: Performance comparison of proprietary and open-source MLLMs on InforgraphicVQA.

| Model | Overall | Description Generation | | | | VQA |
| | | Semantic Consistency | Text Recognition | Detail Coverage | Hallucination Control | Reasoning Ability |
|---|---|---|---|---|---|---|
| InternVL3-78B | 82.59 | 83.47 | 91.71 | 73.14 | 85.32 | 79.31 |
| Qwen2.5-VL-72B | 84.62 | 85.54 | 93.72 | 76.36 | 84.16 | 83.32 |
| Gemini-2.5-pro | 84.49 | 83.47 | 93.47 | 74.67 | 88.98 | 81.87 |
| Claude-4-sonnet | 86.17 | 84.57 | 96.42 | 77.34 | 89.76 | 82.76 |
| GPT-4o | 83.97 | 79.57 | 94.31 | 77.27 | 88.21 | 80.49 |
| GPT-5 | 85.70 | 83.54 | 95.43 | 76.97 | 88.87 | 83.67 |

{*description*}

</LLM Description>

Below is the OCR model's output:

<OCR Output>

{*OCR_text*}

</OCR Output>

During the calibration process, compare the LLM's description with the OCR text, and use the context from the OCR text to identify any parts of the LLM's description, such as locations, people, or objects that are inconsistent with the OCR text. Modify only the inconsistent parts in the LLM's description. Be careful not to change the logical structure of the LLM's output description.

First, briefly explain your analysis and reasoning process, then output the revised content. The output format should be as follows:

<Thought>

Provide a detailed explanation of your reasoning process, showing how you compared and integrated the description content to arrive at the correct description.

</Thought>

<Corrected Result>

Write the corrected content here.

</Calibration Result>

---

**Prompt to summarize multiple corrected descriptions (Description Summarization Agent).**

**User:** Your task is to analyze and compare multiple descriptions from large language models for a 'One-Image Guide' image, and summarize the correct description. Below are the descriptions provided by multiple large language models, each starting with 'Description i:'.

<Description Collection>

{*description*}

</Description Collection>

Some of these descriptions are correct, while others contain errors. Please follow the steps below:

1. Compare and synthesize the multiple descriptions, identifying similarities and differences in their content.

2. Content that is consistent across most descriptions can be considered correct and should be retained.

3. If a model's output clearly deviates from the others, it can be considered incorrect.

4. First, briefly explain your analysis and reasoning process, pointing out the correct and incorrect content in each description, and then output a complete, correct description.

The output format should be as follows:

<Thought>

Provide a detailed explanation of your reasoning process, showing how you compared the descriptions to arrive at the correct one.

</Thought>

<Description>

Write the complete, correct description here.

</Description>

---

**Prompt to generate visual question and answer pairs.**

**User:** Your task is to create 3-5 high-difficulty reasoning-based single-choice questions based on the provided <Reference Text>.

Requirements:

1. The questions must be based on the content of the <Reference Text>, but cannot directly copy the original text. They should require reasoning, summarization, or multi-step logical analysis to arrive at the answer.

2. Each question must have 4 options (A, B, C, D), with only one correct option, and you must provide the correct answer.

3. Do not directly give the answer in the question, and do not include obvious hints in the options.

4. The question, options, and answer should be separated by — on the same line in the following format:

Question content— A. xxx, B. xxx, C. xxx, D. xxx — Correct answer

Each question should occupy one line.

Here is the Reference Text:

<Reference Text>: {*description*}

---

**Prompt to answer the question with a given image.**

**User:** <Image>. You are a visual question answering system. You are given an image, a question, and several options. Carefully observe the content of the image and reason out the option that best matches the facts.

Note:

1. Base your answer only on the information in the image and the question; do not introduce external knowledge.

2. There is only one correct option.

3. Only output the answer (A, B, C, or D); do not output anything else, including your reasoning process.

<Question>

{*question*}

</Question>

<Options>

{*choice*}

</Options>

**Prompt to evaluate the generated description.**

**User:** You are an "One-Image Guide" image description analysis expert. You will be given two inputs:

Ground Truth: The correct "one-image guide" description manually extracted from the image.

Model Prediction: The description generated by the model for the same image.

Your task:

1. Carefully compare the Ground Truth and the Model Prediction.

2. Identify the errors in the Model Prediction and classify them into the following four categories (note: each error can only be assigned to the single most appropriate category, and must not be counted in multiple categories):

- Text Recognition Accuracy: Errors in recognizing text from the image, such as OCR mistakes, spelling errors, etc. Any spelling difference counts as an error, regardless of whether it affects overall understanding.

- Detail Coverage: Missing important details present in the Ground Truth, such as missing locations, activities, attributes, or contextual information. If details are only simplified, replaced with synonyms, or merged in the description, but the core information remains, it is not considered missing. Only when a detail is completely absent or replaced with irrelevant information should it be considered missing.

- Hallucination Control: Content generated that does not exist in the Ground Truth, such as invented locations, events, details, etc. Whether it is a variation, extension, or complete fabrication, it belongs to this category.

- Semantic Consistency: Inconsistencies with the Ground Truth in logic, sequence, location, time, relationships, etc. Minor descriptive differences (synonyms, rhetorical changes) should not affect scoring. Only when the difference causes factual errors, logical conflicts, sequence reversal, or errors in location/time/quantity should points be deducted.

Scoring Rules: (0–1 points):

1.0 point: Completely correct, no errors at all.

0.8 point: Only very few minor errors, not affecting overall understanding.

0.6 point: A certain number of moderate errors, affecting partial understanding.

0.4 point: Many errors, seriously affecting understanding.

0.2 point: Very many errors, almost impossible to understand correctly.

0.0 point: Completely wrong or irrelevant to the Ground Truth.

Output format requirement (must strictly follow the JSON structure below) :

{ "Text Recognition Accuracy": { "Error Description": "Detailed description of errors in text recognition.", "Score": 0-1 }, "Detail Coverage": { "Error Description": "Detailed description of missing details.", "Score": 0-1 }, "Hallucination Control": { "Error Description": "Detailed description of fabricated content.", "Score": 0-1 }, "Semantic Consistency": { "Error Description": "Detailed description of semantic consistency errors.", "Score": 0-1 } }

Considering that OIG-Bench is a bilingual dataset, for images containing Chinese text or originating from Chinese sources, all associated prompts were translated into Chinese, and the evaluated models were instructed to generate responses in Chinese to ensure linguistic consistency between the input and output.

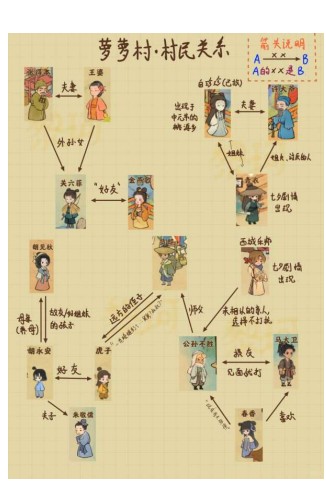

**GPT-4o Description**

Zhang Huaidong and Wang Po are husband and wife, and they have a granddaughter named Guan Liufei. Guan Liufei and Jin Yange are good friends. Zizhenzhen (deceased) and Grandpa Xu were husband and wife, and they had a daughter named Xu Chenzhen. Xu Chuan and Ah Kuan are brothers and sisters. Ah Kuan and a teacher from the Western Regions who was born in Qixi Theater are good friends. He Lang, a teacher from the Western Regions and a member of the Qixi Drama Club, is a sister and brother. Helang and Gongsun Wusheng are martial brothers, while Gongsun Wusheng and Matthew Wei are bad friends, and they will fight each other when they meet. Matthew likes the fragrance of spring. Chunxiang is the disciple of Gongsun Wuying. Huzi and Lang Jianqiu are good friends, and Lang Jianqiu is Lang Yong'an's mother (foster mother). Lang Yong'an and Zhu Jingru are husband and wife.

**Text recognition accuracy:{**
Error description: "There were multiple spelling or recognition errors in the character names predicted by the model. For example, 'Zhang Huaidong' should be 'Zhang Deben', 'Zizhenzhen' should be 'Bai Zhenzhen', 'Matthew Wei' should be 'Ma Dawei', 'He Lang' should be 'Huolang', 'Xiyu Teacher' should be 'Xiyu Musician', and 'Zhu Jingru' should be 'Fuzi (Zhu Jingru). In addition, names such as 'Xu Chenchen 'and' Akuan 'did not appear in Ground Truth, which may be misidentification or fiction. There is also 'Chunxiang is a disciple of Gongsun', which should be 'good sisters/companion'. These are all issues related to the accuracy of text recognition. ",
Rating: 2
},
**Detail coverage:{**
Error description: "The model misses many important details: Bai Zhenzhen's appearance in Taoyuan Township on Zhongyuan Festival, the relationship between Qingyi and Uncle Xu's brother-in-law/annoying person, Qingyi's appearance in the Qixi story, Lang Yong'an's friendship with Huzi and related descriptions (such as" I'm a little annoyed with him now! "," people outside the city... dress strangely "), the goods man is Huzi's distant nephew, Gongsun Zui and Chunxiang are good sisters/companions, arrows at the top right corner of the relationship map, etc. Some teacher-student relationships, and the time of the story are not covered.",
Rating: 2
},
**Hallucination control:{**
Error description: "In the model prediction, many characters and relationships are fictionalized: 'Xu Chuanchuang', 'Ah Kuan', 'Western teacher from Qixi Drama Club', 'He Lang from Qixi Drama Club', 'He Lang and Gongsun Kui are brothers ',' Chun Xiang is Gongsun Kui's apprentice ',' Lang Yongan and Zhu Jingru are husband and wife ', etc. are not shown in Ground Truth, which belongs to fiction. ",
Rating: 1
},
**Semantic consistency:{**
Error description: "The model is seriously inconsistent in terms of character relationship, logical order, kinship, etc. For example, Zhang Huaidong and Wang Po are husband and wife (Zhang Deben), Xu Chuan and Ah Kuan are sister brother relationships, Zhu Jingru and Lang Yong'an are husband and wife (teachers and students), Chunxiang is Gongsun Qiong's apprentice (a good sisters/companion), He Lang and Gongsun Qiong are teacher brothers (no such relationship), Hu Zi and Lang Jianqiu are friends (Lang Yong'an and Hu Zi should be friends), etc. Multiple relationships are confused, resulting in a serious deviation from the original picture in the overall semantic understanding.",
Rating: 1
}

Figure 11: Error case analysis. Note that the original images and model outputs are in Chinese; for ease of presentation, they are translated into English here.

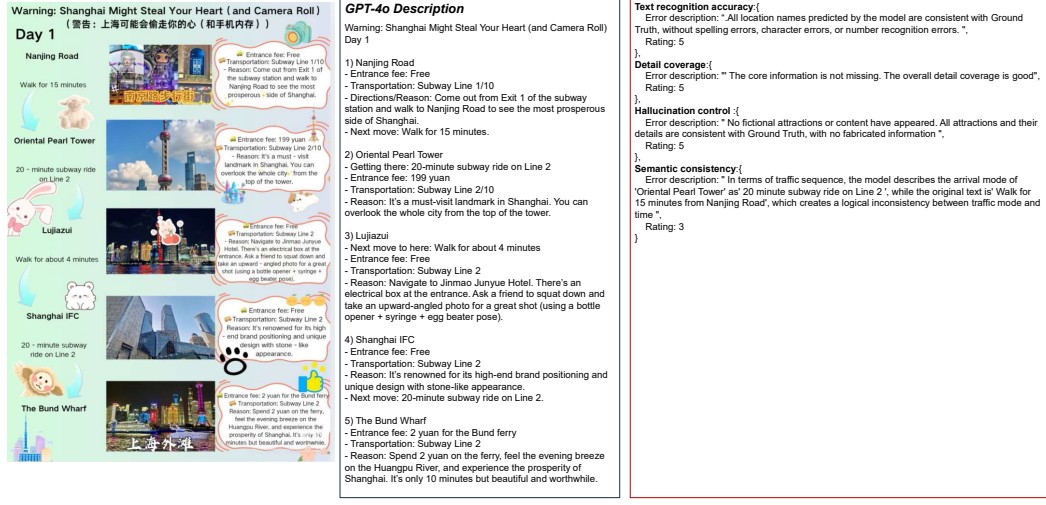

Figure 12: Error case analysis.

## A.9 Additional Error Cases

Figures 11 and 12 present two additional error case analyses. Figure 11 illustrates that when the font style of the game task name is highly stylized, the model's OCR capability drops significantly. Moreover, when the number of arrows is large, the model's perception of relationships becomes weaker, leading to highly confused character relationship recognition. In Figure 12, the error shown in one scenario is that the arrows and textual descriptions are mismatched: the image describes traveling from the Oriental Pearl Tower to Lujiazui by metro, whereas the model incorrectly recognizes the route as arriving at the Oriental Pearl Tower by metro.

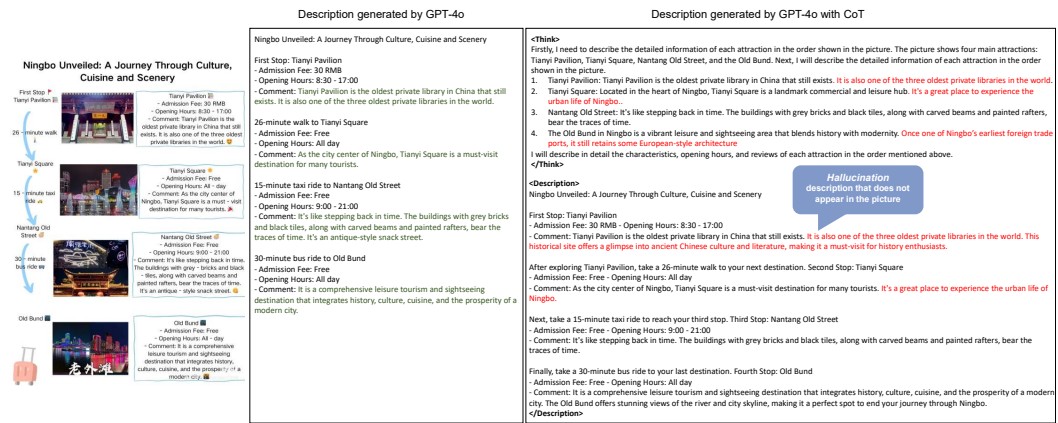

Figure 13: Error case analysis of CoT-augmented output.

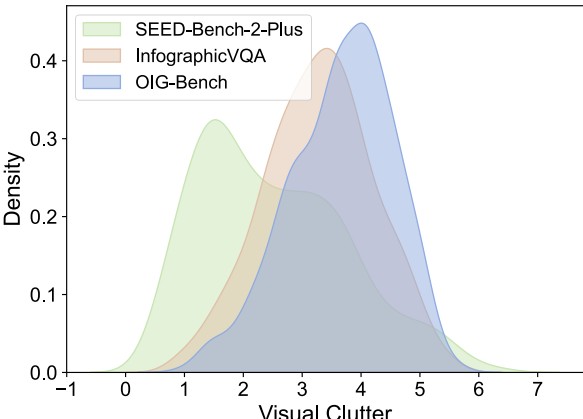

Figure 14: Kernel density estimation of visual clutter of images across OIG-Bench, SEED-Bench2-Plus, and InfographicVQA.

## A.10 MORE DATASET ANALYSIS

To provide a more comprehensive assessment, we additionally consider **visual clutter** Rosenholtz et al. (2007), a widely used theoretical basis for quantifying visual complexity. Visual clutter models the human visual system's sensitivity to the local covariance of color, luminance, and orientation. This approach captures not just how many elements an image contains, but how different they are from each other. A higher visual clutter score indicates greater image complexity. Here. we compute the visual clutter of OIG-Bench, InfographicVQA, and SEED-Bench-2-Plus. As shown in Figure 14, OIG-Bench exhibits consistently higher visual clutter scores compared to the other datasets, indicating that its images are more visually complex and potentially more challenging for both modal perception and reasoning.

We additionally calculated the average output length and maximum output length of each MLLM in the image description generation task. Results are listed in Table 11.

## A.11 LLM USAGE STATEMENT

We used a large language model to help improve the clarity, grammar, and readability of the manuscript. The authors carefully reviewed and edited all LLM-assisted text to ensure accuracy.

Table 11: Statistics of average and maximum description lengths for different MLLMs.

| Model | Avg Words | Max Words |
|---|---|---|
| Groud Truth | 698.89 | 2634.00 |
| Multi-agent System | 665.54 | 2775.00 |
| gpt-5 | 451.94 | 2204.00 |
| gpt-4o | 573.41 | 2836.00 |
| gemini-1.5-pro | 373.80 | 2287.00 |
| claude-4-sonnet-20250514 | 520.55 | 2434.00 |
| claude-3-7-sonnet-20250219 | 516.17 | 2239.00 |
| InternVL3-38B | 593.10 | 2369.00 |
| InternVL3-14B | 626.73 | 2456.00 |
| InternVL3-9B | 630.75 | 2897.00 |
| InternVL2.5-26B | 653.43 | 3513.00 |
| InternVL2.5-8B | 829.26 | 4699.00 |
| InternVL2-26B | 884.74 | 4869.00 |
| Qwen2.5-VL-32B | 1273.85 | 3665.00 |
| Qwen2.5-VL-7B | 611.59 | 3345.00 |
| Qwen2-VL-7B | 569.68 | 3018.00 |
| Qwen2-VL-72B | 593.97 | 3124.00 |
| Qwen2.5-VL-72B | 676.76 | 2875.00 |
| InternVL2_5-38B | 640.52 | 3271.00 |
| InternVL2_5-78B | 743.79 | 2991.00 |
| InternVL2-40B | 836.18 | 3760.00 |
| InternVL3-78B | 710.09 | 2749.00 |
| llava-1.5-7b-hf | 397.43 | 2249.00 |
| llava-1.5-13b-hf | 294.41 | 1880.00 |
| llava-v1.6-mistral-7b-hf | 547.35 | 2235.00 |
| llava-v1.6-vicuna-13b-hf | 443.31 | 2114.00 |

Table 12: Distribution (%) of semantic consistency scores for different models.

| Model | 0 | 0.2 | 0.4 | 0.6 | 0.8 | 1.0 |
|---|---|---|---|---|---|---|
| GPT-4o | 2.88% | 11.97% | 43.01% | 23.50% | 9.75% | 8.86% |
| GPT-5 | 2.35% | 31.80% | 30.45% | 15.20% | 13.50% | 6.70% |
| Qwen2.5-VL-72B | 0.00% | 4.45% | 25.13% | 29.84% | 18.32% | 22.25% |

Table 13: Fine-grained evaluation of semantic consistency in logic diagrams, showing the error distribution.

| Model | Incorrect relation direction | Object recognition error | Misalignment between text and visual cues |
|---|---|---|---|
| Qwen2.5-VL-72B | 41.3% | 12.3% | 31.1% |
| GPT-5 | 52.6% | 11.6% | 34.9% |
| GPT-4o | 61.9% | 15.4% | 36.8% |

Table 14: Comparison of model performance under the Figure+Question and Caption+Figure+Question settings.

| Model | Figure+Question | Caption+Figure+Question |
|---|---|---|
| Qwen2.5-VL-72B | 0.696 | 0.838 |
| GPT-5 | 0.688 | 0.859 |
| GPT-4o | 0.674 | 0.826 |

