# OpenReview forum: "OIG-Bench: A Multi-Agent Annotated Benchmark for Multimodal One-Image Guides Understanding"
_ICLR.cc/2026/Conference — Submitted to ICLR 2026_

### Official Review · Reviewer_cy4B · 2025-10-28

**Soundness:** 2
**Presentation:** 3
**Contribution:** 3
**Rating:** 4
**Confidence:** 5

**Summary:**

This paper presents OIG-Bench, the first benchmark specifically designed to evaluate Multimodal Large Language Models (MLLMs) in understanding "One-Image Guides"—a visual information format that integrates text, images, and symbols to present content in a human-readable, structured manner. The authors develop a semi-automatic annotation pipeline using a multi-agent system to generate high-quality image descriptions, reducing manual labeling costs while ensuring accuracy. They conduct a comprehensive evaluation of 29 state-of-the-art MLLMs (including both commercial and open-source models) on two tasks: description generation and visual question answering (VQA), assessing five key capabilities: semantic consistency, detail coverage, hallucination control, reasoning ability, and text recognition. Results indicate that Qwen2.5-VL-72B performs best overall, but all models exhibit significant shortcomings in semantic understanding and logical reasoning. The proposed multi-agent system outperforms individual MLLMs in description generation, demonstrating its potential as a tool for high-quality dataset construction.

**Strengths:**

Originality: The introduction and formalization of the "One-Image Guide" (OIG) as a distinct and challenging domain for MLLM evaluation is a key contribution. The proposed semi-automatic, multi-agent annotation pipeline can obtain high-quality, detailed annotations

Significance: This paper provides a more realistic and demanding testbed than many existing benchmarks, by focusing on a format that is ubiquitous in real-life (guides, infographics, tutorials) yet challenging for models. The multi-agent annotation pipeline offers a scalable method for generating high-quality data.

Clarity: The paper is well-structured and clear.

**Weaknesses:**

1. In the introduction, the authors mention that “To achieve a complete and accurate understanding of one-image guides, MLLMs must possess fine-grained visual perception (e.g., text and details recognition) and spatial structure parsing capabilities (e.g., interpreting layouts, arrows, and segmented regions).” However, there is no detailed evaluation dimension for spatial parsing ability, such as arrows or segmentation, and it seems that the evaluation could be more comprehensive.

2. The evaluation of cross-modal reasoning capabilities could be more comprehensive. The authors claim that “Experimental results show that current MLLMs still exhibit limitations in semantic understanding and logical reasoning.” However, semantic understanding and logical reasoning seem to have a causal relationship. If a model achieves good semantic understanding of an image (i.e., converting the image into an equivalent textual representation), is it possible that its logical reasoning ability could actually be strong? Which specific dimensions of semantic understanding hinder logical reasoning? Which dimensions still lead to reasoning errors even when semantic understanding is relatively good? The evaluation could be made more fine-grained.

3. The paper does not provide an ablation study to dissect the contribution of each agent. Report the performance (on the same description generation metrics) of: a. A single best model (e.g., GPT-4o alone). b. Multiple models without OCR correction. c. Multiple models with OCR correction but without the final summarization agent (e.g., just picking the "best" description). d. The full pipeline.

**Questions:**

1. Figure 5(b) is unclear, so it is not possible to tell how the ground-truth answer was obtained.

2. Lacks key analysis of OIG-Bench, such as the average/max number of words in the description generated by the model, the average/max number of words in the ground-truth description, and the numbers of single-choice questions and multiple-choice questions.

3. It seems that there is only a reasoning prompt for single-choice questions (“Prompt to answer the question with a given image”), while a reasoning prompt for multiple-choice questions is missing. What is the performance difference between single-choice and multiple-choice questions? What conclusions can be drawn?

4. What is the exact procedure for training and conducting manual annotation? Who performed the manual annotation? What was the specific cost of the manual annotation?

5. Lacks experiments on performance comparison between English and Chinese.

**Details Of Ethics Concerns:**

Nothing

---

> ### Author Response · Authors · 2025-11-22
> **Response to cy4B (part 1/2)**
>
> Thank you for your insightful review and for recognizing the novelty of our work. Below, we provide our point‑to‑point responses to the comments.
>
> > **Weakness 1:** Evalution of spatial parsing ablitly
>
> We thank the reviewer for this insightful comment. The spatial‑parsing elements mentioned in the introduction, such as arrows, segmented regions, and layout cues, are indeed important for understanding one‑image guides. In our benchmark, these factors are incorporated into the **semantic consistency** evaluation dimension.  Descriptions that correctly interpret arrows, region divisions, or layout‑dependent structures are considered semantically consistent, while failures in spatial parsing directly lead to semantic inconsistency. It is worth noting that not all images in our dataset contain such spatial elements; therefore, introducing a separate spatial‑parsing category would not be uniformly applicable across all samples. For this reason, we chose to integrate spatial‑parsing assessment into the broader semantic consistency dimension.
>
> > **Weakness 2:**  Relationship between semantic understanding and logical reasoning
>
> We thank the reviewer for this insightful suggestion. Semantic understanding and logical reasoning are closely related, but they are not equivalent. Semantic understanding provides the factual basis for reasoning, whereas logical reasoning additionally requires multi‑step inference, relationship composition, and task‑specific logical or numerical operations. To better disentangle these two capabilities, we have conducted a fine‑grained analysis.
>
> Specifically, we computed the semantic consistency scores of images corresponding to correct and incorrect answers in VQA for three top-performing models: GPT‑4o, GPT‑5, and Qwen2.5‑VL‑72B. The results are shown in the table below and in **Table 2** in the revised manuscript:
>
> | Model           | Senmatic consistency score when model answers correctly in VQA | Senmatic consistency score when model answers incorrectly in VQA | p-value       |
> |-----------------|------------------------------------------|---------------------------------------------|---------------|
> | gpt-4o          | **64.77**                                    | 59.49                                       | 6*10e-5     |
> | gpt-5           | **65.82**                                    | 53.04                                       | 1*10e-9     |
> | qwen2.5-vl-72b  | **72.14**                                    | 65.75                                       | 2*10e-5    |
>
> These results indicate a significant correlation between semantic understanding and logical reasoning: all models achieve notably higher semantic consistency scores when their VQA answers are correct compared to when they are incorrect. Then, we further analyze the semantic consistency score distribution for images where VQA answers were incorrect. The results are shown in the table below:
>
> | Model           | 0      | 0.2       | 0.4       | 0.6       | 0.8       | 1.0       |
> |-----------------|--------|---------|---------|---------|---------|---------|
> | gpt-4o          | 2.88%  | 11.97%  | 43.01%  | 23.50%  | 9.75%   | 8.86%   |
> | gpt-5           | 2.35%  | 31.80%  | 30.45%  | 15.20%  | 13.50%  | 6.70%   |
> | qwen2.5-vl-72b  | 0.00%     | 4.45%   | 25.13%  | 29.84%  | 18.32%  | 22.25%  |
>
> We found that, in some cases, the semantic consistency score was relatively high (>= 0.8), yet the model still failed to provide the correct VQA answer. This phenomenon is particularly pronounced in the Qwen2.5‑VL‑72B model, indicating that Qwen2.5‑VL‑72B may lack the reasoning capability to fully leverage its semantic understanding in certain cases.
>
> Overall, these results indicate that performing logical reasoning over visual content is a challenging task, in which both semantic understanding and multiple reasoning chains jointly influence the final outcome.

---

> > ### Author Response · Authors · 2025-11-22
> > **Response to cy4B (part 2/2)**
> >
> > > **Weakness 3:**  Ablation study
> >
> > Following your suggestion, we have added an ablation experiment. Specifically, we conducted the following settings:
> > + Removing the entire OCR module (Multi-agent System w/o OCR)
> > + Removing the summarization module and relying solely on a single best model (Qwen2.5‑VL‑72B) with OCR correction (Multi‑agent System w/o Summarize)
> > The results are presented below and in **Table 3** in the revised manuscript:
> >
> > |Model|Overall|Semantic Consistency|Text Recognition Accuracy|Detail Coverage|Hallucination Detection|
> > |-|-|-|-|-|-|
> > |Qwen2.5-VL-72B|79.75|70.48|93.02|70.07|85.42|
> > |Multi-agent System w/o OCR|86.09|77.41|93.36|84.41|89.19|
> > |Multi-agent System w/o Summarize|80.53|71.23|95.41|70.51|84.97|
> > |Multi-agent System|**90.96**|**86.49**|**96.85**|**86.67**|**93.83**|
> >
> > As shown in the table, when removing the OCR module (Multi‑agent System w/o OCR), the text recognition accuracy drops considerably, and semantic consistency also degrades. This is because inconsistent or incorrect OCR outputs across agents lead to misaligned textual references, ultimately harming the global semantic alignment in the final annotations. When removing the summarization module, although the text recognition accuracy improves, the scores for semantic consistency and hallucination detection drop significantly. This indicates that the summarization stage is essential for integrating multi‑perspective signals, mitigating model‑specific biases, and consolidating globally coherent descriptions that go beyond the viewpoint of any single model.
> >
> > > **Question 1:** Unclear of Figure 5b
> >
> > We have replaced Figure 5(b) with a higher-resolution version to improve clarity. We also provide an explanation of how the ground-truth answer was obtained: since there is sufficient wood and oak resin, the key step is to calculate how many copper ingots and iron ingots can be produced. The production materials are shown in the small right corner of the picture. Given that there are only four pieces of coal, it is possible to produce two copper ingots and two iron ingots, which in turn allows for the crafting of at most two barrels.
> >
> > > **Question 2:** Lacks key analysis of OIG-Bench
> >
> > Thank you for the insightful suggestion. We agree that providing a more detailed analysis of OIG-Bench can help readers better understand the characteristics of the dataset and the evaluation setting. In the revised version, we have added comprehensive statistics in **Table 11**. We also provide a representative subset of these statistics below to further illustrate the distributional properties of the dataset.
> >
> > |Model|Avg Words|Max Words|
> > |-|-|-|
> > |Groud Truth|698.89|2634.00|
> > |Multi-agent System|665.54|2775.00|
> > |gpt-5|451.94|2204.00|
> > |gpt-4o|573.41|2836.00|
> > |gemini-1.5-pro|373.80|2287.00|
> > |claude-4-sonnet|520.55|2434.00|
> > |claude-3-7-sonnet|516.17|2239.00|
> > |InternVL3-38B|593.10|2369.00|
> > |Qwen2.5-VL-72B|676.76|2875.00|
> > |Qwen2.5-VL-32B|1273.85|3665.00|
> > |Qwen2-VL-72B|593.97|3124.00|
> >
> > > **Question 3:** Performance difference between single-choice and multiple-choice questions
> >
> > We would like to clarify that OIG‑Bench contains only single‑choice questions. We have corrected all related statements in the paper to avoid misunderstanding.
> >
> > > **Question 4:** Detail of manual annotation
> >
> > Thank you for raising this question. We provide the full details of the training and manual‑annotation procedure below.
> >
> > Manual annotation focuses on validating the correctness and completeness of the ground‑truth descriptions and the automatically generated questions. We employed three annotators, each with prior experience in vision‑language annotation. During annotation, each item was independently labeled by two annotators, and the third annotator served as an adjudicator to determine the final label. This process ensures both consistency and reliability. Note that None of them were involved in model development, which avoids potential bias. The total cost is approximately 60 person-hours.
> >
> > > **Question 5:** Lacks experiments on performance comparison between English and Chinese.
> >
> > We have added this comparison in **Figure 4 and 10** of the revised manuscript. We also list the results below:
> >
> > |Model|Language|Semantic Consistency|Text Recognition Accuracy|Detail Coverage|Hallucination Detection|
> > |-|-|-|-|-|-|
> > |Qwen2.5-VL-72B|Chinese|68.60|92.12|68.61|85.42|
> > ||English|72.93|94.19|71.98|85.87|
> > |GPT-5|Chinese|61.57|92.02|66.59|87.91|
> > ||English|69.37|94.39|69.14|88.67|
> >
> >
> > As shown, Chinese‑language images are generally more difficult. Across both languages, Qwen2.5‑VL‑72B consistently achieves the strongest performance, with a notably large margin in Chinese semantic consistency. One likely reason is that the Qwen2.5‑VL family has been trained on a substantially larger volume of Chinese multimodal data, which enhances its robustness in Chinese multimodal understanding tasks.

---

> > > ### Comment · Reviewer_cy4B · 2025-11-27
> > > **Comprehensiveness of Evaluation**
> > >
> > > Thank you for your reply. I responded from the perspective of improving the article. I feel sorry, my final concern is the comprehensiveness of the evaluation dimensions. For a benchmark in a specific domain, it should have more fine-grained evaluation dimensions than a general model to obtain more helpful conclusions for model training. This should be the significance of a benchmark in a specific domain.
> > >
> > > Suppose I want to train a model to improve this task, what conclusions can I draw from this benchmark? "The conclusion that both semantic consistency and inference contribute to poor results?" or "The conclusion that inference ability is weaker than semantic consistency ability?" The coarse-grained evaluation dimensions in the article are not much different from those of general models. General benchmarks seem to reach similar conclusions; in fact, I would know this conclusion even without testing.
> > >
> > > The conclusions that should be obtained after a benchmark evaluation in a specific domain might be (for example): 1. If I input "image caption + inference question" and the answer is correct, but only input "inference question" and the answer is incorrect, then the inference is not the problem; the problem lies in image perception. Specifically, which semantic consistency dimension is weak, causing my model to perform poorly? 2. My input of "image caption + reasoning question" is also incorrect. So where exactly did it get stuck?

---

> > > > ### Author Response · Authors · 2025-12-01
> > > > **Response to cy4B**
> > > >
> > > > We sincerely thank you for your constructive comments. We fully agree that a domain‑specific benchmark should ideally provide more fine‑grained evaluation dimensions than general benchmarks, as this can yield more actionable insights for model development and targeted training.
> > > >
> > > > In the current version, we adopt relatively coarse‑grained dimensions mainly to ensure evaluation reliability and general applicability across diverse images, thereby providing a stable and trustworthy assessment framework for future research. However, we acknowledge that such coarse‑grained measurements are less informative for identifying the root causes of model errors
> > > >
> > > > To provide more fine-grained evaluation dimensions and more insightful conclusions, we focus our detailed statistical analysis on logic diagrams, which are unique to our benchmark. Logic diagrams typically contain complex arrows, spatial layouts, and directional flows, making them particularly challenging for current models.
> > > >
> > > > We then decompose the evaluation into specific sub-dimensions for semantic consistency of logic diagrams. Specifically, we examine object recognition, spatial relationship understanding, and misalignment between text and visual cues. Results are presented in Table 13 and below:
> > > >
> > > > | Model             | Incorrect relation direction | Object recognition error | Misalignment between text and visual cues |
> > > > |-------------------|------------------------------|---------------------------|-------------------------------------------|
> > > > | Qwen2.5-VL-72B    | 41.3%                        | 12.3%                     | 31.1%                                     |
> > > > | GPT-5    | 52.6%                        | 11.6%                     | 34.9%                                     |
> > > > | GPT-4o             | 61.9%                        | 15.4%                     | 36.8%                                     |
> > > >
> > > > As shown, we observed that incorrect interpretation of arrow relations and misrecognition of text located near arrows constitute the most frequent error types in logic diagram tasks. These results suggest that **current MLLMs show substantial limitations in modeling fine‑grained visual–text relationships, especially those involving directional or structural cues.**
> > > >
> > > > To further investigate the relationship between semantic understanding and logical reasoning, we conduct an additional experiment comparing the "caption+figure+question" setting with the original "figure+question" setting, as the reviewer suggested. Results are presented below and in Appendix Table 14 in the revised manuscript.
> > > >
> > > > | Model            | Figure+Question | Caption+Figure+Question |
> > > >  |------------------|--------------------------|-----------------|
> > > >  | Qwen2.5-VL-72B   | 0.696                    | 0.838           |
> > > >  | GPT-5            | 0.688                    | 0.859           |
> > > >  | GPT-4o           | 0.674                    | 0.826           |
> > > >
> > > > | Caption+Figure+Question | Figure+Question | GPT-5 | GPT-4o | Qwen2.5-VL-72B |
> > > >  |-------------------------|-----------------|--------|--------|------------------|
> > > >  | √    | √               | 65.5%  | 62.7%  | 64.9%           |
> > > >  | √    | ×               | 20.4%  | 23.2%  | 19.0%           |
> > > >  | ×        | √               | 3.3%   | 4.7%   | 4.8%            |
> > > >  | ×                       | ×               | 10.6%  | 9.4%   | 11.3%           |
> > > >
> > > > As shown, when the ground‑truth captions are provided, the VQA performance improves significantly, indicating that **the bottleneck lies in semantic understanding rather than reasoning.**
> > > >
> > > > Finally, to better illustrate the value of our benchmark, we summarize and present our principal insights as follows:
> > > >
> > > > 1. **Performance gap between open- and closed-source MLLM is narrowing.** Several open‑source models are able to match or even surpass closed‑source models. In particular, Qwen2.5‑VL‑72B achieves the highest overall score among all evaluated models, outperforming the best closed‑source model, GPT‑5, by 2.6%
> > > >
> > > > 2. **MLLMs exhibit strong text recognition ability but struggle with semantic consistency.** Most models reach around 80% accuracy in text recognition, but their performance on semantic consistency drops to about 60%.  In particular, the semantic understanding of arrow‑based relations is especially weak.
> > > >
> > > > 3. **Weak semantic understanding substantially hinders logical reasoning.** MLLMs exhibit significantly higher semantic consistency scores when their VQA answers are correct compared to when they are incorrect. When providing correct image descriptions as auxiliary information, the logical reasoning score of the model also significantly improves.
> > > >
> > > > 4. **Semantic understanding capability improves as model scale increases.** In the Qwen family from 7B to 72B, we see consistent improvements across five capabilities, with the largest gains in semantic consistency and hallucination control.
> > > >
> > > > All these findings have been incorporated into Section 4.1 of the revised manuscript.

---

### Official Review · Reviewer_RCyg · 2025-10-30

**Soundness:** 3
**Presentation:** 2
**Contribution:** 3
**Rating:** 4
**Confidence:** 4

**Summary:**

Aiming at the problem that the understanding ability of multimodal large language models (MLLMs) in the "One-Image Guide" is insufficient, this paper proposes the first benchmark dataset OIG-Bench focusing on this scenario, and designs a multi-agent semi-automatic labeling process to reduce the construction cost. At the same time, 29 mainstream MLLMs are systematically evaluated. The research motivation is clear, and the paper fills the evaluation gap of "human-like cognitive level single-graph guide understanding". Data screening combines MLLM cross-validation and manual review to evaluate the core capabilities of dimension coverage and human verification.

**Strengths:**

1. Existing multimodal benchmarks mostly focus on diagrams/documents of layout rules, while "single-graph guide", as a human-friendly visual format that integrates text, images, and symbols (such as travel route maps, game guide maps), is more close to the complex information presentation of real scenes, but there is a lack of specific evaluation benchmarks before. This paper takes this scene as the research object for the first time, and clarifies its "human-like cognitive characteristics" and the three major challenges faced by MLLMs: fine grain visual details, complex logical structures, and cross-modal reasoning. The research scene is innovative and practical.
2. The evaluation plan is well designed. The image description generation task covers "text recognition, semantic consistency, detail coverage, illusion control", and the VQA task focuses on "reasoning ability", which are the core capabilities of single-image guide understanding.
3. Evaluate 29 MLLMs (including 5 closed-source and 24 open-source ones, covering scales from 6B to 78B), with unified prompt templates, temperature=0, and other experimental settings, to conduct a fair comparison of the performance of different models, across domains (games/travel/medical/food), and graph types (layout diagrams/logic diagrams).

**Weaknesses:**

1. The document mentions that the data is collected from Xiaohongshu, Baidu, and Google, but only states that "anonymization ensures privacy," and does not mention whether it is authorized by the platform or uses "publicly available and copyright-free content," which poses a risk of infringement. It is recommended to supplement the "Data Copyright Compliance Statement" to clarify that the data source is publicly available for non-commercial use, or has been authorized by the platform for research.
2. Existing experiments only demonstrate the overall performance of multi-agents (e.g., 86.4% semantic consistency, 96.8% text recognition), and do not validate the necessity of each component of "description-generated agent, OCR-generated agent, OCR-corrected agent, summary agent". It is recommended to supplement ablation experiments, such as the change in semantic consistency after removing OCR-corrected agents, and the performance difference between agents and multi-agents generated using only a single description, to strengthen the persuasiveness of the advantages of multi-agent collaboration.
3. The existing analysis only compares "image entropy" and fails to compare the performance of OIG-Bench with existing benchmarks on the same model, making it difficult to intuitively demonstrate the unique value of OIG-Bench. It is recommended to select multiple representative models, reproduce and evaluate them on the "description generation" and "VQA" tasks of InfographicVQA, compare the score differences between OIG-Bench and InfographicVQA, quantify the challenges of OIG-Bench, and strengthen the argument of "filling the evaluation gap".

**Questions:**

The OIG-Bench benchmark and multi-agent labeling method proposed in this paper provide a high-quality tool for the evaluation of "single-graph guide understanding" of MLLMs, and the experimental results have important guiding significance for model optimization and practical application. If the above ablation experiments and compliance instructions can be supplemented, the scientificity, integrity and persuasiveness of the paper will be further enhanced.

---

> ### Author Response · Authors · 2025-11-22
> **Response to RCyg**
>
> We sincerely thank you for recognizing the novelty and practicality of our work and for your insightful suggestions. Below, we provide our point‑to‑point responses to the comments.
>
> > **Weakness 1:** Data copyright compliance statement
>
> We thank the reviewer for raising this important point. All data used in OIG‑Bench are collected exclusively from publicly accessible web pages and openly available materials on Xiaohongshu, Baidu, and Google. We include only content that is publicly viewable without login restrictions and is permitted for non‑commercial research use. In addition, all images undergo strict anonymization, and no identifiable personal information is retained. We will further add a “Data Copyright Compliance Statement” to the paper to explicitly clarify the data sources, usage permissions, and our compliance with non‑commercial research and fair‑use principles.
>
> > **Weakness 2:** Ablation study missing for the multi-agent system
>
> To verify the necessity of each component in our multi‑agent pipeline, we have added an ablation experiment. Specifically, we conducted the following settings:
> + Removing the entire OCR module (**Multi-agent System w/o OCR**)
> + Removing the summarization module and relying solely on a single model (Qwen2.5‑VL‑72B) with OCR correction (**Multi‑agent System w/o Summarize**)
> The results are presented below and in **Table 3** in the revised manuscript:
>
> |Model|Overall|Semantic Consistency|Text Recognition Accuracy|Detail Coverage|Hallucination Detection|
> |-|-|-|-|-|-|
> |Qwen2.5-VL-72B|79.75|70.48|93.02|70.07|85.42|
> |Multi-agent System w/o OCR|86.09|77.41|93.36|84.41|89.19|
> |Multi-agent System w/o Summarize|80.53|71.23|95.41|70.51|84.97|
> |Multi-agent System|**90.96**|**86.49**|**96.85**|**86.67**|**93.83**|
>
> As shown in the table, when removing the OCR module (Multi‑agent System w/o OCR), the text recognition accuracy drops considerably, and semantic consistency also degrades. This is because inconsistent or incorrect OCR outputs across agents lead to misaligned textual references, ultimately harming the global semantic alignment in the final annotations. When removing the summarization module, although the text recognition accuracy improves, the scores for semantic consistency and hallucination detection drop significantly. This indicates that the summarization stage is essential for integrating multi‑perspective signals, mitigating model‑specific biases, and consolidating globally coherent descriptions that go beyond the viewpoint of any single model
>
> > **Weakness 3:**  Benchmark evaluation on InfographicVQA
>
> We thank the reviewer for the valuable suggestion. To more intuitively demonstrate the unique value of OIG‑Bench, we additionally evaluate several representative models on both the “image description” and “VQA” tasks of InfographicVQA. Due to the time constraints of the rebuttal period, we randomly sample 300 images from InfographicVQA, and apply the same annotation, verification, and evaluation procedures as in OIG‑Bench.
>
> The results are presented below and in **Table 10** in the revised manuscript:
>
> |Model|Overall|Semantic Consistency|Text Recognition|Detail Coverage|Hallucination Control|Reasoning Ability|
> |-|-|-|-|-|-|-|
> |InternVL3-78B|82.59|83.47|91.71|73.14|85.32|79.31|
> |Qwen2.5-VL-72B|84.62|**85.54**|93.72|76.36|84.16|83.32|
> |Gemini-2.5-pro|84.49|83.47|93.47|74.67|88.98|81.87|
> |Claude-4-sonnet|**86.17**|84.57|**96.42**|**77.34**|**89.76**|82.76|
> |GPT-4o|83.97|79.57|94.31|77.27|88.21|80.49|
> |GPT-5|85.70|83.54|95.43|76.97|88.87|**83.67**|
>
> The results show that model scores on InfographicVQA are consistently higher than their corresponding scores on OIG‑Bench, indicating that OIG‑Bench presents a significantly greater challenge. This comparison quantitatively highlights the difficulty of OIG‑Bench and supports our claim that it fills an important evaluation gap in assessing fine‑grained semantic understanding and reasoning for one‑image guides.

---

### Official Review · Reviewer_1T1E · 2025-10-31

**Soundness:** 3
**Presentation:** 3
**Contribution:** 3
**Rating:** 6
**Confidence:** 4

**Summary:**

This paper introduces an evaluation benchmark for MLLMs called OIG-Bench, which focuses on one-image guide understanding. To build this benchmark, the authors employ a multi-agent-based semi-automated construction method that effectively generates image descriptions and reduces manual annotation costs. Evaluation of 29 mainstream MLLMs across five dimensions of one-image guide understanding shows that existing models still have substantial room for improvement in semantic understanding and logical reasoning.

**Strengths:**

- The paper is well-written and easy to follow, with clear visualizations.
- A more complex benchmark on infographics is constructed, whose data contain richer visual logical and textual content.
- An effective multi-agent architecture is proposed for dataset construction, which achieves strong results across all metrics, and can benefit future works.

**Weaknesses:**

- Multi-option bias: The multi-agent system generates options based on descriptions. Could this introduce bias in MLLMs if the target option distribution is not uniform, especially without post-processing such as shuffling?
- Complexity metrics: Image entropy may not sufficiently reflect structural complexity. For example, a full-text image could also yield high image entropy.
- Need more examples: In line 465, "One possible explanation is that…" would be more convincing with additional failure cases.
- Some typos: For instance, in line 300, "log" -> "\log".

**Questions:**

- Could the multi-agent system introduce bias in the target option distribution during the construction of benchmark, if no post-processing (such as shuffling) is applied?
- Although overly simple images are filtered out, the complexity of the remaining data is not clearly explained. How is the difficulty of the generated questions or options evaluated?
- Image entropy may not be a convincing metric for structural complexity. Are there potentially better evaluation metrics?

---

> ### Author Response · Authors · 2025-11-22
> **Response to 1T1E**
>
> We greatly appreciate the very detailed feedback and your recognition of our contributions. We hope our response below effectively addresses your concerns.
>
> > **Weakness 1 and Question 1:** Multi-option bias.
>
> To address your concern, we performed two additional analyses. First, we examined the option distribution on OIG‑Bench. We found that **the proportions of options A, B, C, and D are 20.7%, 38.6%, 35.9%, and 4.8%, respectively**, showing that the generated options are indeed more concentrated in A and B, while D appears least frequently.
>
> To improve robustness, we adopted the **CircularEval** strategy used in MMBench[1], which evaluates each question across all circular permutations of the answer options. **We re‑evaluated the VQA results under this setting, and updated the scores in Table 1**.  The table below presents the VQA results before and after applying CircularEval:
>
> | Model               | Original | CircularEval |
> |---------------------|----------|--------------|
> | GPT‑5               | 65.13    | 55.26        |
> | Claude‑4‑sonnet     | 59.14    | 53.55        |
> | Gemini‑1.5‑pro      | 63.30    | 55.62        |
> | Qwen2.5‑VL‑72B      | 68.83    | 63.62        |
> | InternVL3.5‑38B     | 63.49    | 55.72        |
>
> We found that the scores become noticeably lower under CircularEval. This result further highlights the insufficient logical reasoning capabilities of current MLLM models.
>
> > **Weakness 2 and Question 3:** Complexity metrics
>
> We acknowledge that image entropy alone may not sufficiently capture structural complexity. To provide a more comprehensive assessment, we additionally consider **visual clutter** [2], a widely used theoretical basis for quantifying visual complexity. Visual clutter models the human visual system’s sensitivity to the local covariance of color, luminance, and orientation. This approach captures not just how many elements an image contains, but how different they are from each other. A higher visual clutter score indicates greater image complexity. Following [2], we compute the visual clutter of OIG-Bench, InfographicVQA, and SEED-Bench-2-Plus. Results are presented in **Figure 14** of the revised manuscript and discussed in **Appendix A.10**. We also listed the average visual clutter across different benchmarks below:
>
> |Dataset|Average Visual Clutter|
> |-|--|
> |OIG‑Bench|**3.67**|
> |InfographicVQA|3.28|
> |SEED‑Bench‑2‑Plus|2.52|
>
> These results show that OIG‑Bench exhibits consistently higher visual clutter scores compared to the other datasets, indicating that its images are more visually complex and potentially more challenging for both modal perception and reasoning.
>
> > **Weakness 3:** More case of CoT
>
> To make our explanation more convincing, we have added an illustrative failure case in **Figure 13** in the appendix of the revised manuscript. This case shows the differences in image descriptions before and after applying the CoT prompt.  As shown, the CoT‑augmented output does not exhibit any improvement compared to the baseline description. In fact, the additional reasoning steps introduced by the CoT prompt sometimes lead to unnecessary elaboration or speculative content, which can reduce factual accuracy. This observation suggests that incorporating CoT prompting does not provide tangible benefits for image description generation in our benchmark setting.
>
> > **Weakness 4:** Typo
>
> Thanks for pointing out this typo. We have corrected it in the revised manuscript.
>
> > **Question 2:** The complexity of OIG-Bench is not clearly explained
>
> To address this concern, we conducted an evaluation on the InfographicVQA dataset using the same procedures as in OIG‑Bench. Specifically, due to time and cost constraints, we randomly sampled 300 images from InfographicVQA and applied the same annotation, verification, and evaluation pipeline as used for OIG‑Bench. The results are shown in the table below and in **Table 10** in the revised manuscript.
>
> |Model|Overall|Semantic Consistency|Text Recognition|Detail Coverage|Hallucination Control|Reasoning Ability|
> |-|-|-|-|-|-|-|
> |InternVL3-78B|82.59|83.47|91.71|73.14|85.32|79.31|
> |Qwen2.5-VL-72B|84.62|**85.54**|93.72|76.36|84.16|83.32|
> |Gemini-2.5-pro|84.49|83.47|93.47|74.67|88.98|81.87|
> |Claude-4-sonnet|**86.17**|84.57|**96.42**|**77.34**|**89.76**|82.76|
> |GPT-4o|83.97|79.57|94.31|77.27|88.21|80.49|
> |GPT-5|85.70|83.54|95.43|76.97|88.87|**83.67**|
>
> The results show that model scores on InfographicVQA are consistently higher than their corresponding scores on OIG‑Bench, indicating that OIG‑Bench presents a significantly greater challenge. This comparison quantitatively highlights the difficulty of OIG‑Bench.
>
>
> [1] Liu Y, Duan H, Zhang Y, et al. Mmbench: Is your multi-modal model an all-around player?[C]//European conference on computer vision. Cham: Springer Nature Switzerland, 2024: 216-233.
>
> [2]  Rosenholtz R, Li Y, Nakano L. Measuring visual clutter[J]. Journal of Vision, 2007, 7(2): 17-17.

---

### Official Review · Reviewer_DjWo · 2025-10-31

**Soundness:** 3
**Presentation:** 3
**Contribution:** 3
**Rating:** 4
**Confidence:** 4

**Summary:**

This paper addresses the gap in evaluating MLLMs’ human-like understanding of **One-Image Guides (OIGs)**—text-imagery-symbol composites optimized for human cognition—by proposing **OIG-Bench**, the first dedicated benchmark. Key contributions include: (1) OIG-Bench itself: 808 bilingual (56.6% Chinese, 43.4% English) OIGs across 4 domains (game, travel, food, medicine) with 2800 reasoning questions, featuring higher visual complexity (average entropy 5.8) than existing benchmarks (e.g., InfographicVQA: 4.8). (2) A **semi-automated multi-agent annotation pipeline**: specialized agents (description generation via GPT-4o/Claude-4, OCR via PaddleOCR, correction/summarization via GPT-4.1) collaborate to generate high-quality labels, reducing manual effort and outperforming single models. (3) Comprehensive evaluation of 29 MLLMs (6B–78B, proprietary/open-source): Qwen2.5-VL-72B achieves the best overall accuracy (77%), but all models struggle with semantic consistency and logical reasoning. The paper also reveals domain/type disparities (game/travel harder than food/medicine; logic diagrams harder than layout) and prompt trade-offs (CoT aids VQA but harms descriptions).

**Strengths:**

* The paper identifies a clear and valuable gap in MLLM evaluation. "One-Image Guides" are a distinct, commonly used format that is more complex and less rigidly structured than standard charts or documents. The quantitative analysis showing OIG-Bench images have a higher average image entropy than related benchmarks (InfographicVQA, SEED-Bench-2-Plus) supports the claim of higher visual complexity.
- The paper thoroughly evaluates its framework against relevant and strong baselines, including "Strong-to-Weak Distillation" and "Multi-Agent Collaboration". The method is tested across a wide array of 11+ benchmarks covering comprehensive understanding, hallucination, chart/table understanding, and knowledge-oriented tasks, demonstrating broad improvements.
- The paper provides a thorough evaluation of 29 state-of-the-art MLLMs, including both open-source and proprietary models. The findings are insightful, pinpointing a clear weakness across all models: they perform well on "Text Recognition" but "struggle to accurately interpret complex visual-text relationships".

**Weaknesses:**

* The evaluation for both tasks (Description Generation and VQA) relies heavily on GPT-4.1 as the "Judge Model". While the authors provide a human correlation study (Table 5, Fig. 8) for two models , this reliance is a potential source of bias. The judge model's own limitations or stylistic preferences could unfairly penalize or reward certain models, and its ability to robustly evaluate 27 other models for complex reasoning is not fully established.
* While OIG-Bench is bilingual, there is no analysis of model performance disparities between Chinese and English OIGs.
* The benchmark contains 808 images. While the filtering process is rigorous, this is a relatively small dataset. Furthermore, the domain distribution is skewed, with "Game" (31.2%) and "Travel" (30%) accounting for over 60% of the data. The paper notes that models perform worst in these two domains, which means the overall performance scores are heavily weighted by these challenging, but potentially niche, domains.
* The authors need to provide more reasoning MLLMs to support experimental conclusions, such as Gemini-2.5-pro and o3.
* The data and code not be publicly released.

**Questions:**

* The multi-agent annotation system (Section 3.1.2) uses GPT-4.1 for both the "OCR Correction Agent" and the "Description Summarize Agent". The evaluation (Section 3.2) also uses GPT-4.1 as the primary judge model. How do you account for the potential bias where the evaluation judge (GPT-4.1) might favor the stylistic, structural, or reasoning patterns of the same model family used in the data annotation pipeline?
* Do models perform differently on Chinese vs. English OIGs? If yes, is the gap due to OCR accuracy (e.g., worse English OCR), language model proficiency, or annotation differences?

I look forward to an active discussion with the authors during the rebuttal phase and will revise my score accordingly.

---

> ### Author Response · Authors · 2025-11-22
> **Response to DjWo (part 1/2)**
>
> Thank you for the detailed and constructive feedback! We treasure the opportunity to address your concerns and improve our work. Below, we provide our point‑to‑point responses to the comments.
>
> > **Weakness 1:** The evaluation using GPT-4.1 is a potential source of bias.
>
> To further validate the robustness of our evaluation, we conducted additional experiments using two extra judge models: **Gemini 2.5 Pro** and **Claude‑4‑Sonnet**. The evaluation follows the exact evaluation pipeline outlined in our paper. We have reevaluated the top-10 models in Table 1. The detailed results are presented below and in **Appendix Tables 8 and 9**  of the revised manuscript,  and further discussed in **Appendix A.4**.
>
> **Gemini‑2.5‑pro‑based Evaluation:**
> |Model|Overall|Semantic Consistency|Text Recognition Accuracy|Detail Coverage|Hallucination Detection|
> |-|-|-|-|-|-|
> |InternVL3-78B|64.26|45.24|84.12|55.35|72.34|
> |Qwen2-VL-72B|64.52|46.75|87.88|52.84|72.60|
> |Qwen2.5-VL-7B|62.99|40.90|_88.89_|49.57|70.62|
> |Qwen2.5-VL-32B|_68.41_|53.05|**89.32**|**59.86**|71.41|
> |Qwen2.5-VL-72B|**70.13**|**58.17**|85.71|_58.75_|77.90|
> |Gemini-1.5-pro|64.22|50.27|80.94|49.97|74.12|
> |Gemini-2.5-pro|64.22|52.23|81.22|52.19|74.23|
> |GPT-4o|67.04|50.67|83.43|53.67|**82.38**|
> |o3|65.86|_53.41_|81.67|50.79|77.57|
> |GPT-5|67.51|51.75|86.12|52.13|_80.03_|
>
> **Claude‑4‑Sonnet-based Evaluation:**
> |Model|Overall|Semantic Consistency|Text Recognition Accuracy|Detail Coverage|Hallucination Detection|
> |-|-|-|-|-|-|
> |InternVL3-78B|66.94|58.21|77.23|71.35|60.98|
> |Qwen2-VL-72B|67.54|57.36|78.64|67.69|66.45|
> |Qwen2.5-VL-7B|64.34|51.85|78.83|63.12|63.56|
> |Qwen2.5-VL-32B|71.32|64.60|_80.84_|**74.46**|65.37|
> |Qwen2.5-VL-72B|**77.45**|**72.27**|**84.88**|_74.28_|80.35|
> |Gemini-1.5-pro|67.38|60.72|76.66|62.37|71.75|
> |Gemini-2.5-pro|73.27|_66.38_|79.64|70.96|78.11|
> |GPT-4o|70.53|63.17|67.88|66.71|**82.38**|
> |o3|_74.51_|64.12|78.98|71.97|_80.97_|
> |GPT-5|71.77|63.44|80.60|63.78|79.25|
>
> The Spearman correlation for the rank of these models between the GPT‑4.1‑based OIG-Bench and the Gemini‑2.5‑pro‑based OIG‑Bench is **0.884 with p-value<0.05**, and between the GPT‑4.1‑based OIG‑Bench and the Claude‑4‑Sonnet‑based OIG‑Bench is **0.903 with p-value<0.05**. This consistency reflects that the inherent bias introduced by GPT-4.1 in OIG-Bench is not significant.
>
> This consistency stems from the evaluation design: the judge model compares the MLLM output with the ground‑truth text, focusing on semantic alignment. This is a task modern LLMs perform with high stability and reliability[1,2]. Thus, despite stylistic differences among judge models, outcomes are driven primarily by objective semantic consistency rather than subjective variations.
>
> To make the evaluation more rigorous and transparent, we plan to provide multiple versions of benchmark results in the final release:
> 1. Single‑judge evaluation using GPT‑4.1 only (as in the current paper).
> 2. Multi‑judge averaged evaluation using GPT‑4.1, Gemini 2.5 Pro, and Claude‑4.5‑Sonnet.
>
> Users can choose different evaluation strategies according to their cost constraints or stability requirements,  thereby enabling more flexible benchmarking practices.
>
> [1] Zheng L, Chiang W L, Sheng Y, et al. Judging llm-as-a-judge with mt-bench and chatbot arena[J]. Advances in neural information processing systems, 2023, 36: 46595-46623.
>
> [2] Bavaresco A, Bernardi R, Bertolazzi L, et al. Llms instead of human judges? a large scale empirical study across 20 nlp evaluation tasks[C]//Proceedings of the 63rd Annual Meeting of the Association for Computational Linguistics (Volume 2: Short Papers). 2025: 238-255.
>
> > **Weakness 2 and Question 2:** Language analysis of OIG-Bench
>
> We have added this comparison in **Figure 4 and 10** of the revised manuscript. We also list the results below:
>
> |Model|Language|Semantic Consistency|Text Recognition Accuracy|Detail Coverage|Hallucination Detection|
> |-|-|-|-|-|-|
> |Qwen2.5-VL-72B|Chinese|68.60|92.12|68.61|85.42|
> ||English|72.93|94.19|71.98|85.87|
> |GPT-5|Chinese|61.57|92.02|66.59|87.91|
> ||English|69.37|94.39|69.14|88.67|
>
> As shown, Chinese‑language images are generally more difficult. Across both languages, Qwen2.5‑VL‑72B consistently achieves the strongest performance, with a notably large margin in **Chinese semantic consistency**. One likely reason is that the Qwen2.5‑VL family has been trained on a substantially larger volume of Chinese multimodal data, which enhances its robustness in Chinese multimodal understanding tasks.

---

> > ### Author Response · Authors · 2025-11-22
> > **Response to DjWo (part 2/2)**
> >
> > > **Weakness 3:** Domain distribution is skewed
> >
> > We sincerely thank the reviewer for the insightful comments on the dataset size and domain distribution. Although OIG‑Bench comprises 808 images, its construction follows a rigorous multi‑stage filtering pipeline combined with extensive human verification to ensure annotation accuracy and semantic consistency. Such stringent quality control inevitably constrains the dataset size, but it also enhances its challenge level and representativeness for evaluating multimodal models.
> >
> > Regarding the domain distribution, the imbalance is not manually imposed but instead reflects the natural prevalence of one‑image‑guide content on real platforms. **In the raw data source, the Game and Travel domains account for a large proportion, aligning with their high‑frequency usage in actual one‑image‑guide scenarios**. For example, the Game domain is commonly used for sharing gameplay tips, character information, and event summaries, while the Travel domain frequently contains trip itineraries, attraction highlights, and travel recommendations.
> >
> > While these domains may appear “niche” in some contexts, they are in fact among the most active and widely used categories in real one‑image‑guide applications. Moreover, images in these domains often exhibit complex layouts, dense textual content, and rich semantic structures, posing substantial challenges for OCR and multimodal reasoning.
> >
> > In future work, we plan to expand the dataset and further refine the domain coverage, while maintaining the same strict quality standards.
> >
> > > **Weakness 4:** More reasoning MLLMs
> >
> >  We have added the results of additional reasoning‑oriented MLLMs, including **Gemini‑2.5‑Pro** and **o3**, to **Table 1** in the revised manuscript. We also list the results below:
> >
> > |Model|Overall|Semantic Consistency|Text Recognition Accuracy|Detail Coverage|Hallucination Detection|Reasoning Ability|
> > |-|-|--|-|-|-|-|
> > |Gemini-2.5-Pro|72.30|67.12|89.18|66.32|83.07|55.81|
> > |o3|71.15|63.36|84.15|65.14|85.24|57.82|
> >
> > As shown, these reasoning models still have limitations in semantic understanding and logical reasoning on OIGs.
> >
> > > **Weakness 5:** Data and code not be publicly released
> >
> > We would like to clarify that an anonymous GitHub link has already been provided at the end of the Introduction in the original manuscript, and we will publicly release the data used in our experiments on HuggingFace, along with the corresponding evaluation code on GitHub, upon acceptance of the paper.
> >
> > > **Question 1:** Model bias of GPT-4.1
> >
> > To address this, we conducted an ablation study, in which GPT‑4.1 originally used for the OCR Correction Agent and Description Summarize Agent was replaced with two alternative models: Gemini 2.5 Pro and Claude‑4‑Sonnet. The evaluation stage remained unchanged, with GPT‑4.1 serving as the judge model.
> >
> > The results are presented below and in **Table 3** in the revised manuscript:
> >
> > |Model|Overall|Semantic Consistency|Text Recognition Accuracy|Detail Coverage|Hallucination Detection|
> > |-|-|-|-|-|-|
> > |Qwen2.5-VL-72B|79.75|70.48|93.02|70.07|85.42|
> > |Multi-agent System (Claude‑4‑Sonnet)|90.38|85.79|96.14|85.98|93.61|
> > |Multi-agent System (Gemini-2.5-Pro)|90.20|85.34|**97.03**|84.13|**94.31**|
> > |Multi-agent System (GPT-4.1)|**90.96**|**86.49**|96.85|**86.67**|93.83|
> >
> > As shown, the multi‑agent annotations generated using these alternative models still achieve consistently high evaluation scores across all dimensions, and their quality remains higher than that produced by any single best-performing MLLM. This demonstrates that the superiority of the multi‑agent pipeline does not stem from GPT‑4.1’s stylistic or structural preference for annotations produced by its own model family, but rather from the inherent advantages of the multi‑agent design itself.

---

### Author Response · Authors · 2025-12-01
**Summary of Paper Revision**

Dear Program Chairs, Senior Area Chairs, Area Chairs, and Reviewers:

We sincerely thank you for your time and expertise, and we are grateful to all reviewers — DjWo, 1T1E, RCyg, and cy4B — for their constructive insights. We are glad to find that most reviewers acknowledged our contributions (e.g., DjWo: OIG-Bench identifies a clear and valuable gap in MLLM evaluation. The findings are insightful; RCyg: The research scene is innovative and practical; cy4B: This paper provides a more realistic and demanding testbed.)

To further strengthen the validity and robustness of our work, we conducted a series of additional experiments and analyses:

> **``` Reducing evaluation bias```**

1. To enhance the robustness of the evaluation process and reduce potential model bias, we introduced Gemini 2.5 Pro and Claude‑4‑Sonnet as additional judge models and repeated the entire evaluation pipeline on the top‑10 models. The results are reported in **Tables 8 and 9** and discussed in **Appendix A.4**. The rankings produced by different judge models remain highly correlated.

2. We further replaced GPT‑4.1 with these two models in the OCR Correction Agent and Description Summarization Agent. As shown in **Table 3**, the performance remains consistent, confirming the robustness of our proposed multi‑agent annotation system.

3. In addition, we adopted CircularEval to eliminate option bias and updated all VQA scores accordingly in **Table 1**.

> **``` Validating the multi‑agent pipeline```**

1. To verify the necessity of each component in our multi‑agent annotation system, we conducted additional ablations. As the results shown in **Table 3** and discussed in **section 4.3**, removing each individual agent leads to clear performance degradation, demonstrating the essential role of each component in ensuring annotation quality and stability.

> **``` Demonstrating benchmark challenge```**

1. To further demonstrate the challenge posed by our benchmark, we computed the visual clutter to more comprehensively quantify image complexity. Results are presented in **Figure 14** and discussed in **Appendix A.10**.

2. In addition, we evaluated several representative models on 300 randomly sampled InfographicVQA images using the same pipeline as OIG‑Bench. As shown in **Table 10** and discussed in **Appendix A.7**, the models consistently achieve higher scores on InfographicVQA, confirming that OIG‑Bench is substantially more challenging and fills a critical gap in evaluating fine‑grained semantic reasoning for one‑image guides.

> **``` Providing deeper insights```**

Following reviewer cy4B's suggestion, to deepen our analysis, we conducted additional fine‑grained analyses of our benchmark. Firstly, we decomposed the semantic‑consistency evaluation, with results shown in **Table 13**. We also examined the relationship between semantic understanding and logical reasoning, as presented in **Table 2** and **Table 14**. Together with our previous analyses, our results lead to several key insights:

+ Open‑source models are rapidly catching up.

+ Models recognize text well but struggle with semantic consistency, particularly in arrow‑based relations.

+ Weak semantic understanding substantially limits logical reasoning.

+ Semantic understanding improves with model scale.

We deeply appreciate the significant time and effort that ACs devote during this particularly demanding period. We hope that these additional analyses and clarified experimental results can be fully considered in your final decision. Thank you again for your careful handling of our submission.

---

### Meta-Review · Area_Chair_2Hce · 2026-01-05

**Summary:**

This paper proposes a benchmark for measuring the capability of multimodal large language models (MLLMs), One-Image Guides (OIG); OIG is a visual structure containing text, image, and symbols. The proposed OIG-Bench contains 808 bilingual OIGS (56.6% Chinese and 43.4% English) across 4 domains (game, travel, food, and medicine) with 2800 questions. This paper shows that the selected images are more complex than images from other benchmarks based on image entropy comparison. The data generation pipeline is based on multiple commercial high-performing MLLMs by generating image descriptions (using GPT4o, Claude-4-Sonnet, Gemini-1.5-Pro, and Qwen2.5-VL-78B), OCR generation (using PaddleOCR), OCR correction (using GPT-4.1), and description summarizing (using GPT-4.1). 29 MLLMs are compared on the proposed benchmark. This paper shows that OIG is still a challenging task, especially in semantic understanding and logical reasoning.

**Reviewer Concerns:**

This paper has borderline initial recommendations, leaning towards negative. The main concerns raised by the reviewers can be summarized as follows.

First, there were some concerns, addressed well by the rebuttal comment

- Needs more reasoning MLLMs (DjWo)
    - The rebuttal comments show additional experiments with Gemini‑2.5‑Pro and GPT-o3
- Image entropy may not sufficiently reflect structural complexity (1T1E)
    - Additional image visual clutter results are shown
- Potential risk of the existance of multi-option bias (1T1E)
    - Additional results with a circularEval strategy are provided.
- Potential legal issues, such as data copyright / data publicity (DjWo, RCyg)
    - Although the rebuttal comment clarified that (1) they will release the code and dataset (and the current version is available at the anonymous github link), and (2) all images are publicly viewable without login restrictions and are permitted for non-commercial research use. I double checked whether screenshots can be used in an academic publication, and it seems that this is a case of a fair use, hence there is no significant problem: [link](https://dl.digra.org/public/journals/2/whitepaper1.pdf)
- Chinese vs. English analysis (DjWo, cy4B)

However, the following concerns still remain after the discussion period.

- The benchmark contains only 808 samples, which is a very small number of images (DjWo)
- The domain distribution is skewed with game (31.2%) and travel (30%). This could be problematic because the paper notes that these two domains are where the models show weak performances (DjWo)
- The comprehensiveness of the evaluation dimensions is unclear (cy4B)

There is a partially addressed concern, but the AC thinks it could be potentially problematic

- For both description generation and VQA, the evaluation heavily relies on GPT-4.1 as the judge model. This could lead to a bias during evaluation (DjWo)
    - The rebuttal comment showed additional results using two extra judge models (Gemini 2.5 Pro and Claude‑4‑Sonnet). The results show that the correlation between ranks is relatively high (0.884 and 0.903 in Spearman correlation). However, this only supports that the current commercial models have a similar rank preference, and this cannot ensure that the future results will be consistent.

Another minor issues that does not affect the decision:

- I found that the data is saved in `xsxl`, rather than widely used data formats, such as csv or json. I personally suggest to use more popular data formats, such as csv, json, txt or tsv.
- While reviewing the paper, I noticed the author's identity from the anonymous github; the author's identity is revealed in the huggingface link on the repository. After discussion with SAC, we decided that this is not a case of desk rejection, but I strongly suggest to anonymitize the image repositories too.


Overall, I think the disadvantage of this paper (the scale of the datatset is too small, the domain distribution is skewed with game and travel, the data construction and evaluation pipelines heavily rely on LLMs) overweighs its advantages. Hence, I recommend rejection for this paper.

**Reviewer Scores:**

In my opinion, some reviewers can bump up their opinions to borderline accept. For example, the major concerns of 1T1E and RCyg seem to be addressed by the rebuttal comment. I don't think their opinions would go above "accept", but they would mildly change their opinions to borderline accept.

However, in my opinion, concerns raised by DjWo and cy4B would not be fully addressed, and their opinions wouldn't be changed above the borderline.

Overall, I think that if there were heavy discussions between the authors and the reviewers, this paper would have borderline opinions.

---

### Decision · Program_Chairs · 2026-01-26

Reject